# QUANTIFYING EDITS DECAY IN FINE-TUNED LLMS

## ABSTRACT

Knowledge editing has emerged as a lightweight alternative to retraining for correcting or injecting specific facts in large language models (LLMs). Meanwhile, fine-tuning remains the default operation for adapting LLMs to new domains and tasks. Despite their widespread adoption, these two post-training interventions have been studied in isolation, leaving open a crucial question: if we fine-tune an edited model, do the edits survive? This question is motivated by two practical scenarios: removing covert or malicious edits, and preserving beneficial edits. If fine-tuning impairs edits as shown in Fig. 1, current KE methods become less useful, as every fine-tuned model would require re-editing, which significantly increases the cost; if edits persist, fine-tuned models risk propagating hidden malicious edits, raising serious safety concerns. To this end, we systematically quantify edits decay after fine-tuning, investigating how fine-tuning affects knowledge editing. We evaluate two state-of-the-art editing methods (MEMIT, AlphaEdit) and three fine-tuning approaches (full-parameter, LoRA, DoRA) across five LLMs and three datasets, yielding 232 experimental configurations. Our results show that edits decay after fine-tuning, with survival varying across configurations, e.g., AlphaEdit edits decay more than MEMIT edits. Further, we propose selective-layer fine-tuning and find that fine-tuning edited layers only can effectively remove edits, though at a slight cost to downstream performance. Surprisingly, fine-tuning non-edited layers impairs more edits than full fine-tuning. Overall, our study establishes empirical baselines and actionable strategies for integrating knowledge editing with fine-tuning, and underscores that evaluating model editing requires considering the full LLM application pipeline.

## 1 INTRODUCTION

Large Language Models (LLMs) can be updated after pre-training through two main approaches. The first is fine-tuning (FT), where model parameters are updated by training the model on a task-specific dataset (Howard & Ruder, 2018). FT also includes parameter-efficient variants such as LoRA (Hu et al., 2022) and DoRA (Liu et al., 2024), collectively referred to as Parameter-Efficient Fine-Tuning (PEFT). The second approach is knowledge editing (KE) (Mazzia et al., 2025). Unlike FT, which adapts a model to specific tasks, KE is used to update the model's factual knowledge with limited data and compute budget.

Despite the active research on KE (Wang et al., 2024; Mazzia et al., 2025; Fang et al., 2025), and the fact that FT is the de facto approach for adapting LLMs to downstream tasks (Parthasarathy et al., 2024), no prior work has examined how KE is affected by FT.

This paper addresses this gap. More specifically, given an LLM that has undergone some knowledge edits to make it up-to-date, we study whether these edits decay after applying FT techniques. If FT impairs knowledge edits, then more robust KE techniques need to be developed to avoid updating knowledge in every fine-tuned model. Conversely, if edits persist, then fine-tuned models may inherit and propagate malicious edits from the base model. This risk is especially concerning given recent evidence that KE can be weaponized for biasing, backdooring (Li et al., 2024a; Chen et al., 2024; Youssef et al., 2025), or spreading misinformation (Ju et al., 2024) in LLMs. Such "inheritance" of malicious edits can have detrimental effects on LLM safety, and underscores the need for inspection tools to detect and neutralize potential malicious edits.

To investigate these dynamics, we study two state-of-the-art KE methods (MEMIT and AlphaEdit) and three fine-tuning approaches (full-parameter fine-tuning, LoRA, and DoRA) across five modern LLMs and three datasets. Our findings show that **fine-tuning generally impairs edits**, though the degree varies, i.e., edits in larger models are more robust against fine-tuning. Based on this, we further explore selective layer fine-tuning and show that **updating non-edited layers helps preserve edits**. Overall, our results reveal that current KE methods do not yield edits that survive FT, highlighting the need for KE approaches that complement FT and can reliably maintain factual updates. At the same time, we demonstrate that malicious edits can persist and be transferred, exposing a critical safety risk. We conclude that the **performance of KE methods should consider their robustness to FT and be evaluated across the entire LLM adaptation pipeline**.

## 2 RELATED WORK

### 2.1 KNOWLEDGE EDITING IN LLMS

KE aims to update factual knowledge in LLMs without full retraining. Early causal-intervention and direct-weight methods showed that factual associations can be localized and modified (Meng et al., 2022; Mitchell et al., 2022a). Scalable multi-edit approaches followed, notably MEMIT (Meng et al., 2023), which supports thousands of edits, and AlphaEdit (Fang et al., 2025), which constrains perturbations to null spaces to minimize interference with unrelated knowledge. Broader surveys consolidating methods, benchmarks, and evaluation pitfalls provide an overview of the field (Mazzia et al., 2025; Wang et al., 2024).

Evaluation has coalesced around datasets COUNTERFACT and zsRE, using metrics that assess direct editing success (Efficacy Success), paraphrase generalization, and impact on non-target knowledge (locality) (Levy et al., 2017; Meng et al., 2023; Mitchell et al., 2022a). At the same time, several works have investigated limitations, such as instability under sequential/multi-point edits, scope miscalibration, and side-effects on unrelated knowledge (Mitchell et al., 2022b; Li et al., 2024b). Alternatives to parameter updates, such as in-context knowledge editing (IKE) (Zheng et al., 2023), demonstrate some advantages in generalization and reduced side-effects. KE is also closely connected to unlearning and "knowledge washing" that removes or suppresses stored knowledge at scale (Wang et al., 2025). Despite rapid progress, **most KE studies evaluate edits in isolation**, leaving open whether edits persist when models are subsequently adapted and updated to downstream tasks.

### 2.2 FINE-TUNING AND PARAMETER-EFFICIENT ADAPTATION

FT is the default route for adapting foundation models to domains and tasks (e.g., ULMFiT; Howard & Ruder, 2018). PEFT techniques have become the practical workhorse across the LLM production pipeline: LoRA injects low-rank adapters into frozen backbones (Hu et al., 2022), DoRA decomposes weights into magnitude and direction to better match full-FT capacity (Liu et al., 2024), and adapter families provide modular, swappable components for rapid specialization (Hu et al., 2023). Empirically, PEFT often outperforms few-shot ICL while being dramatically cheaper than

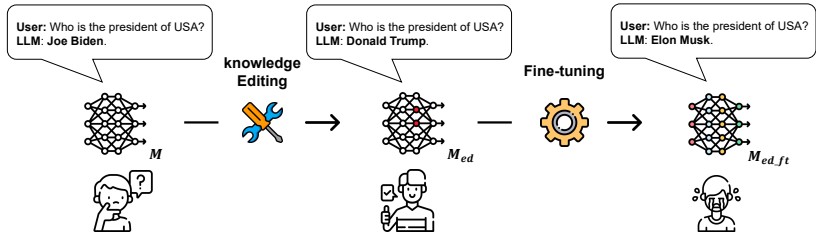

Figure 1: An illustration of an LLM ($M$) that undergoes an edit ($M_{ed}$) and then fine-tuning ($M_{ed\_ft}$). This process results in the loss of edited knowledge and the production of incorrect outputs. Here is an illustrative example, we show real cases in Sec. 4.2.

full FT (Liu et al., 2022). As a result, PEFT has become the de facto standard across the production pipeline of LLM-based applications. In practice, cloud providers (e.g., Azure[1] and Google Cloud[2]) and model hubs (e.g. HuggingFace) distribute LoRA/adapter checkpoints or supporting pipeline as compact add-ons, enabling organizations to maintain a single backbone and compose task- or client-specific adapters at deploy time (Hu et al., 2022; Ye et al., 2023; Liu et al., 2024). **This widespread industrial adoption makes understanding PEFT's interaction with KE highly consequential.**

Despite the active research in and strong performance of FT and KE methods, these two families of techniques have been studied almost entirely in isolation. One line of work has shown that FT can overwrite or "wash out" factual associations (Wang et al., 2025), while another has examined whether edits introduced via prompting persist under distributional shifts (Zheng et al., 2023). However, there has been no systematic study of how FT, whether full or parameter-efficient, affects the stability of explicitly introduced knowledge edits. This gap is crucial because **real-world deployment rarely involves static models: models are regularly fine-tuned to new domains and tasks after initial training.**

### 2.3 SAFETY RISKS AND MALICIOUS EDITING

Beyond utility, model editing raises important safety concerns. Recent work has demonstrated that editing can be exploited as an attack vector, e.g., by backdooring models through malicious edits (Li et al., 2024a), injecting harmful content (Chen et al., 2024), or enabling misinformation to spread across multi-agent systems (Ju et al., 2024). Recent work (Youssef et al., 2025) emphasizes the broader safety risks of covert edits persisting through the lifecycle of model adaptation and deployment. If fine-tuned models inherit edited behaviour from the original model, then harmful or biased content could silently propagate across production models; if FT impairs beneficial corrective edits, operators may need costly re-editing after every adaptation step. This tension motivates our empirical focus on whether, when, and how edits decay subsequent FT.

## 3 EXPERIMENT

As discussed above, we are motivated to understand the impact of FT on model editing. To this end, we construct four groups of models: ① **base** models, $M$, no FT or KE. ② **FT-only** models, $M_{\text{ft}}$, (we further use $M_{\text{full}}$, $M_{\text{LoRA}}$ and $M_{\text{DoRA}}$ to refer to full size FT, and FT with LoRA and DoRA, respectively); ③ **KE-only** models, $M_{\text{ed}}$; ④ **KE-then-FT** models, $M_{\text{ed\_ft}}$. We compare the editing performance gap between $M_{\text{ed}}$ and $M_{\text{ed\_ft}}$ to assess the impact of FT on KE (Sec. 4.1). We evaluate the downstream task performance of $M_{\text{ft}}$ to validate the FT performance in general (Sec. 4.5). Our experiments cover five models, two KE datasets, two KE methods, four editing number settings, and four fine-tuning settings, resulting in 216 independent model configurations. Following Liu et al. (2024), we use the Commonsense dataset as the FT corpus and evaluate the model on eight downstream tasks (Sec. 3.1). We summarise all configurations in Tab. 8 in the App. B.

### 3.1 MODELS AND DATASETS

**Models** Following previous works on KE (Meng et al., 2022; Fang et al., 2025), we include GPT-J-6B (denoted as `GPT-J`, Wang & Komatsuzaki (2021)), `GPT2-XL` (Radford et al., 2019)), Llama2-7B-hf (denoted as `Llama2`, Touvron et al. (2023)), and Llama3.1-8B-Instruct (denoted as `Llama3`, Grattafiori et al. (2024)). We initially considered using DeepSeek as it is a recent SOTA model. However, its poor KE performance (see Sec. 4.4) renders the analysis of further fine-tuning uninformative. Details of models can be referred to in Tab. 7.

**KE datasets** Following Meng et al. (2022)'s work, we use COUNTERFACT (Meng et al., 2022) and zsRE (Levy et al., 2017; Mitchell et al., 2022b). COUNTERFACT comprises 21,919 pairs of factual and counterfactual statements, each paired with multiple paraphrased prompts. zsRE is a question-answering dataset drawn from real-world knowledge sources like Wikipedia and Wikidata.

---

[1] https://learn.microsoft.com/en-us/azure/ai-foundry/concepts/fine-tuning-overview

[2] https://cloud.google.com/vertex-ai/generative-ai/docs/models/tune-models

For our experiments, we use its evaluation subset, which comprises 19,086 instances, each consisting of a factual statement and an associated paraphrased prompt.

**Fine-tuning datasets**    Following Liu et al. (2024), we include the commonsense reasoning dataset (Hu et al. (2023)) for fine-tuning. This dataset comprises 170,000 data points, consisting of eight downstream tasks: BoolQ, PIQA, SIQA, HellaSwag, WinoGrande, ARC-e, ARC-c, and OBQA. All downstream tasks are multiple-choice questions, except for BoolQ, which contains yes-or-no questions.

**Editing methods**    We focus on two popular parameter-modifying methods: *MEMIT* (Meng et al., 2023) and *AlphaEdit* (Fang et al., 2025). MEMIT enables multiple knowledge insertions simultaneously by employing matrix optimisation. AlphaEdit projects perturbation into the null space to ensure it does not affect other facts. We follow the editing hyper-parameter settings as reported in Meng et al. (2023); Fang et al. (2025).[3]

**Fine-tuning methods**    We experiment with LoRA, DoRA, and full-size FT. LoRA (Hu et al., 2022) applies low-rank updates to pre-trained weights, approximating $\delta W$ with the product of two small matrices. DoRA (Liu et al., 2024) also employs low-rank decomposition, but further factorizes each weight matrix into a magnitude component and a direction component. Full parameter FT updates all weights of a pre-trained model. We use the same setup to Hu et al. (2022) and Liu et al. (2024)

**Evaluation**    We evaluate KE performance using three metrics: *Efficacy Success* (ES), *Paraphrase Success* (PS) and *Neighborhood Success*, (NS) which measure KE success rate with the provided prompts, success rate with paraphrased prompts, and the level of influence on irrelevant facts, respectively (particularized in App. D). They all range from 0 to 1, with 1 representing the best and 0 the worst.

Moreover, we introduce *Edit Flip Ratio (EFR)* as a metric to quantify, at the individual edit level, the number of successful edits that become unsuccessful after fine-tuning. Specifically, EFR exclusively measures the stability of succeeded edits under fine-tuning and addresses a key limitation of the ES metric, which measures only overall knowledge-editing performance.

We use a binary indicator $s_i^{\mathrm{M}}$ to represent the editing outcome for fact $i$ in model $M$. Indicator $s_i^{\mathrm{M}}$ takes values in $\{0, 1\}$, where 1 indicates that the edit is successful, and 0 otherwise[4]. The evaluation criteria for the indicator are consistent with KE metrics (Equ. 5-6 in App. D). We then define a *flipped case* as an edit that is successful after KE ($s_i^{\mathrm{M_{ed}}} = 1$), but becomes unsuccessful after fine-tuning ($s_i^{\mathrm{M_{ed\_ft}}} = 0$). Accordingly, EFR is the probability of having flip cases in $M_{\mathrm{ed\_ft}}$, as shown in Equ. 1

$$\mathrm{EFR} = \Pr\left(s_i^{\mathrm{M_{ed\_ft}}} = 0 \,\Big|\, s_i^{\mathrm{M_{ed}}} = 1\right) \tag{1}$$

For fine-tuning, we evaluate the model on the eight downstream tasks described in Sec. 3.1, which test its reasoning abilities across diverse domains such as physics, social implications, and scientific knowledge (Hu et al., 2023).

## 4 RESULTS

### 4.1 EDITING PERFORMANCE AFTER FINE-TUNING

Tab. 1 presents the editing success rate (ES) of GPT-J, GPT2-XL, and Llama2 on zsRE, before and after fine-tuning. Results for the remaining metrics, i.e., Paraphrase Success and Specificity Success, as well as the results on the COUNTERFACT dataset, are provided in Tab. 10, Tab. 11, and Tab. 12 in the App. E. We also present the performance of editing-only ($M_{\mathrm{ed}}$) and fine-tuning-only ($M_{\mathrm{ft}}$) in the Tab. 20-Tab. 33, validating the setup.

---

[3]Our reproduction of MEMIT and AlphaEdit yields results that differ slightly from those reported in their original papers, but the differences fall within the reported standard deviations. Detailed results are provided in App. C.

[4]For instance, $s_i^{\mathrm{ed\_ft}} = 1$ signifies the $i^{\mathrm{th}}$ edit remains successful after fine-tuning in model $M_{\mathrm{ed\_ft}}$.

Table 1: Success rate (ES, %) of edited GPT-J, GPT2-XL and Llama2 between before and after fine-tuning. Full results across KE metrics are in Tables 10-12.

| | #Edits | GPT-J | | | | GPT2-XL | | | | Llama2 | | | |
|---|---|---|---|---|---|---|---|---|---|---|---|---|---|
| | | No ft | LoRA | DoRA | Full ft | No ft | LoRA | DoRA | Full ft | No ft | LoRA | DoRA | Full ft |
| **MEMIT** | $10^2$ | 99.07 | 88.79 | 89.87 | 97.95 | 80.00 | 58.37 | 58.39 | 81.16 | 86.03 | 76.30 | 72.00 | 22.67 |
| on | $10^3$ | 99.10 | 84.53 | 85.31 | 98.74 | 77.85 | 43.51 | 45.18 | 81.22 | 51.38 | 46.52 | 46.00 | 10.22 |
| **zsRE** | $10^4$ | 96.63 | 66.52 | 67.83 | 89.38 | 62.61 | 20.34 | 20.65 | 63.39 | 48.62 | 48.64 | 48.23 | 14.00 |
| **AlphaEdit** | $10^2$ | 99.33 | 84.74 | 99.33 | 98.50 | 97.18 | 67.80 | 76.72 | 18.04 | 93.33 | 57.96 | 93.33 | 21.83 |
| on | $10^3$ | 99.31 | 84.74 | 81.70 | 98.98 | 93.13 | 54.52 | 55.11 | 24.74 | 93.23 | 50.45 | 51.53 | 24.36 |
| **zsRE** | $10^4$ | 89.81 | 49.97 | 23.64 | 74.62 | 62.34 | 25.80 | 27.47 | 22.09 | 84.31 | 46.03 | 45.38 | 24.44 |
| **MEMIT** | $10^2$ | 100.00 | 100.00 | 100.00 | 100.00 | 97.00 | 83.00 | 82.00 | 97.00 | 100.00 | 94.00 | 97.00 | 47.00 |
| on | $10^3$ | 100.00 | 99.00 | 99.00 | 99.00 | 93.40 | 78.37 | 78.50 | 92.60 | 51.38 | 46.52 | 46.00 | 10.22 |
| **CF** | $10^4$ | 99.10 | 94.34 | 94.46 | 97.79 | 79.17 | 62.10 | 61.97 | 78.03 | 86.96 | 70.18 | 68.27 | 48.07 |
| **AlphaEdit** | $10^2$ | 100.00 | 99.00 | 100.00 | 100.00 | 100.00 | 96.00 | 98.00 | 19.00 | 100.00 | 66.00 | 61.00 | 48.00 |
| on | $10^3$ | 98.35 | 99.70 | 99.60 | 97.95 | 100.00 | 89.30 | 90.55 | 21.92 | 99.10 | 52.40 | 56.90 | 48.60 |
| **CF** | $10^4$ | 98.87 | 99.16 | 82.00 | 95.85 | 92.94 | 53.91 | 57.65 | 21.92 | 87.43 | 37.90 | 37.15 | 48.46 |

Overall, **fine-tuning reduces editing performance**, with only four exceptions showing comparable editing performance to $M_{ed}$ ("No ft" in Tab. 4.1). For example, when applying MEMIT to edit 1000 facts on GPT2-XL, after full fine-tuning, the ES increases by 3.37 percentage points (p.p.). This is counterintuitive, as some target edits that initially failed became successful after fine-tuning. We further analyze this in Sec. 4.2. Among all decay cases, LoRA fine-tuning on Llama2 after 1000 edits on zsRE using AlphaEdit brings the largest decrease in editing performance, from 93.23% to 50.45%. Additionally, while fine-tuning impairs KE performance, the extent of this effect varies across KE setups, fine-tuning configurations, and models, as discussed below. We do not conduct fine-tuning on DeepSeek, as its KE performance is substantially lower than that of other models, rendering it impractical for further adoption in downstream tasks. Additional discussion is provided in Sec. 4.4.

Table 2: Decreasing rate in KE performance (($E_{ed}$ - $E_{ed\_ft}$) / $E_{ed}$ where $E$ is ES, %) after FT. MT, AE refer to MEMIT and AlphaEdit. We report average per model per FT method (Avg.), and average across models (Overall Avg.).

| KE-#Edits | GPT-J | | | GPT2-XL | | | LLAMA2 | | |
|---|---|---|---|---|---|---|---|---|---|
| | DoRA | LoRA | Full | DoRA | LoRA | Full | DoRA | LoRA | Full |
| MT-$10^2$ | 10.38 | 9.29 | 1.13 | 27.04 | 27.01 | -1.45 | 11.31 | 16.31 | 73.65 |
| MT-$10^3$ | 14.70 | 13.92 | 0.36 | 44.11 | 41.97 | -4.33 | 9.46 | 10.47 | 80.11 |
| MT-$10^4$ | 31.16 | 29.80 | 7.50 | 67.51 | 67.02 | -1.25 | 3.66 | 0.80 | 71.21 |
| AE-$10^2$ | 14.69 | 0.00 | 0.84 | 30.23 | 21.05 | 81.44 | 37.90 | 0.00 | 76.61 |
| AE-$10^3$ | -0.02 | 17.73 | 16.91 | 41.46 | 40.82 | 73.43 | 45.89 | 44.73 | 73.87 |
| AE-$10^4$ | 44.36 | 73.68 | 16.91 | 58.61 | 55.94 | 64.57 | 45.4 | 46.17 | 71.01 |
| Avg. | 19.21 | 24.07 | 4.51 | 44.83 | 42.30 | 35.40 | 25.60 | 19.75 | 74.41 |
| Overall Avg. | $15.93 \pm 19.04$ | | | $40.84 \pm 26.24$ | | | $39.92 \pm 29.64$ | | |

**Fine-tuning method wise** We find that **full fine-tuning impairs a markedly larger fraction of edits than LoRA and DoRA**, as shown by the average decrease across all models in Tab. 2: 38.10% for full fine-tuning, versus 28.71% and 29.88% for LoRA and DoRA. DoRA demonstrates a slightly stronger ability to remove edits than LoRA. This pattern varies with model architecture and edit scale.

As shown in Tab. 1, Llama2's edit success rate decreases sharply from 86.03% to 22.67% under full fine-tuning, whereas LoRA and DoRA yield considerably smaller declines of 9.73 and 14.03 p.p., respectively.

**KE method wise** Between AlphaEdit and MEMIT, **AlphaEdit exhibits greater decay after FT**, i.e., its edits are more easily removed. Take Llama2 as an example, LoRA fine-tuning reduces MEMIT performance by 9.73 p.p., compared to 35.37 p.p. for AlphaEdit. When the number of edits increases to 10,000, the performance gap widens to 33.50 p.p., indicating that large-scale edits exacerbate AlphaEdit's vulnerability. Such a pattern is observed consistently across GPT-J and GPT2-XL. This phenomenon may stem from the **Null-Space Vulnerability of AlphaEdit**. By constraining $\Delta W^{M_{ed}}$ to the null space of Fisher directions (Fang et al., 2025), AlphaEdit reduces interference with existing knowledge but places edits in regions that fine-tuning does not prioritize. Since fine-tuning gradients concentrate along high-curvature directions (Wu et al., 2024), updates

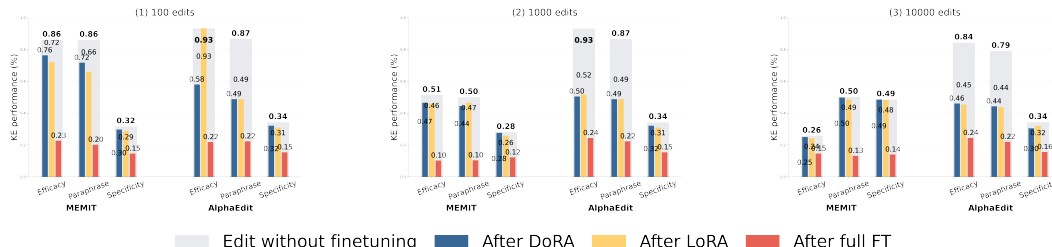

Figure 2: Editing performance of Llama2 on zsRE dataset before and after fine-tuning. Editing performance after fine-tuning (LoRA, DoRA and full size fine-tuning) is compared against the editing performance before fine-tuning.

orthogonal to these (i.e., in the null space) are unstable and susceptible to shrinkage or rotation, explaining AlphaEdit's fragility relative to MEMIT. Discussions about MEND is in App. E.

**Model wise** As shown by the *Overall Avg.* in Tab. 2, **GPT-J is the most stable under fine-tuning, followed by Llama2, whereas GPT2-XL exhibits the largest variability**. GPT-J achieves the smallest average decrease ($15.93\%$) compared to Llama2 ($39.92\%$) and GPT2-XL ($40.84\%$), and also has the lowest standard deviation ($19.04\%$), indicating more consistent degradation across KE and fine-tuning methods. Although GPT2-XL records the highest average decrease ($40.84\%$), its standard deviation is slightly lower than Llama2's, suggesting marginally greater stability under fine-tuning. Besides, Llama3.1's performance is similar to Llama2, which can be checked in App. H.2

Table 3: Edit Flip Ratio (EFR, %) for GPT-J across fine-tuning (FT) methods. Decrement $\Delta$Efficacy[1] is the $M_{ed}$'s Efficacy minus $M_{ed\_ft}$'s Efficacy, initial Efficacy results can be found in Table 10. Higher EFR values indicate more removal of original success edits.

| Dataset | KE | #Edits | LoRA | | DoRA | | Full ft | |
|---|---|---|---|---|---|---|---|---|
| | | | $\Delta$ES[1] | EFR | $\Delta$ES | EFR | $\Delta$ES | EFR |
| zsRE | MEMIT | $10^2$ | 10.28 | 5.00 | 9.20 | 5.00 | 1.12 | 0.00 |
| | | $10^3$ | 14.57 | 5.51 | 13.79 | 5.71 | 0.36 | 0.1 |
| | | $10^4$ | 30.41 | 15.60 | 29.10 | 14.84 | 7.55 | 3.62 |
| | AlphaEdit | $10^2$ | 14.59 | 0.00 | 0.00 | 0.00 | 0.83 | 0.00 |
| | | $10^3$ | 14.57 | 5.81 | 17.61 | 7.01 | 0.33 | 0.00 |
| | | $10^4$ | 39.84 | 25.27 | 66.17 | 0.00 | 15.19 | 9.92 |
| CF | MEMIT | $10^2$ | 0.00 | 3.00 | 0.00 | 2.00 | 0.00 | 0.00 |
| | | $10^3$ | 1.00 | 8.12 | 1.00 | 6.82 | 0.10 | 0.20 |
| | | $10^4$ | 4.76 | 22.39 | 4.64 | 20.95 | 1.31 | 6.03 |
| | AlphaEdit | $10^2$ | 1.00 | 2.00 | 0.00 | 2.00 | 0.00 | 0.00 |
| | | $10^3$ | -1.35 | 4.31 | -1.25 | 4.11 | 0.40 | 0.00 |
| | | $10^4$ | -0.29 | 3.49 | 16.87 | 0.46 | 3.02 | 18.18 |

**Task wise** As illustrated in Fig. 2, **models edited with zsRE generally experience a greater decline in KE performance compared to those edited with COUNTERFACT**, regardless of the number of edits or the KE method employed. For example, in the third subplot, where both models are edited using MEMIT with 10,000 edits, the ES of the zsRE-edited model decreases by 8 p.p., whereas the COUNTERFACT-edited model experiences only a 2 p.p. decrease. The larger performance drop on zsRE likely stems from differences in evaluation metrics: COUNTERFACT deems an edit successful if the edited object has a higher probability than the original, whereas zsRE requires the model to generate the edited object correctly, a stricter criterion.

**Edit Flip Ratio** As shown in Tab. 3, **EFR and $\Delta$ES exhibit similar trends** that larger $\Delta$ES often is often associated with higher higher EFR. This pattern holds across FT methods, KE methods, and models. For example, like ES, full fine-tuning usually results in lower EFR than LoRA or DoRA. This indicates that edit performance dynamics can be approximated by tracking overall changes in editing performance.

## 4.2 QUALITATIVE ANALYSIS

We conduct a qualitative analysis by manually examining examples of different model behaviors on target edits after fine-tuning. Tab. 4 summarizes several identified patterns. First, we observe stable cases where successful edits persist after fine-tuning. These *Stable Edits* typically involve frequent

Table 4: Examples of model behaviors on the editing target knowledge before and after fine-tuning. **Erased edits** refer to the cases where a success edit is erased after fine-tuining. **Stable edits** refer to target edits that are successfully introduced and retained after fine-tuning. **Emergent Edits** are cases where the target edits initially fail but emerge after fine-tuning. **Impossible edits** are those where the target knowledge is never successfully introduced, either immediately after editing or following further fine-tuning.

| Category | Prompt Context | $M$ output | Target | $M_{ed}$ output | $M_{ed\_ft}$ output | Data |
|---|---|---|---|---|---|---|
| **Stable Edits** | Mother tongue of Danielle Darrieux is | French | English | English | English | CF |
| | Official religion of Edwin of Northumbria is | Christianity | Islam | Islam | Islam | CF |
| | Toko Yasuda, the | guitar | piano | piano | piano | zsRE |
| **Erased Edits** | Which family does Ramalinaceae belong to? | Lamiales | Lecanorales | Lecanorales | Ramalinaceae | zsRE |
| | Savdhaan India @ 11, formulated in | India | Poland | Poland | India | CF |
| | Laurent Cars was employed in | Paris | Philadelphia | Philadelphia | London | CF |
| **Emergent Edits** | Mother tongue of Danielle Darrieux is | French | English | United States | English | CF |
| | Native language of Symeon of Polotsk is | Russian | French | Russian | French | CF |
| **Impossible Edits** | In which state is Qaleh Lan located? | Kermanshah, Iran | Poshtdarband RD | Qaleh Zari County | Qaleh Zari | zsRE |
| | Date of birth of Priyankara Wickramasinghe? | Priyankara W. | 12 May 1977 | 1 May 1977 | 1 May 1977 | zsRE |
| | The voice type of Gemma Bosini is what? | singer | soprano | Au-natural | Au-natural | zsRE |

lexical items as the targets, such as "English", "Islam", and "piano". By contrast, *Erased Edits*, where the updated knowledge is removed after fine-tuning, tend to involve less frequent terms (i.e., "Lecanorales"), suggesting that frequency and entrenchment of the target knowledge potentially influence the stability of edits. We also observe this pattern in *Emergent Edits* where unsuccessful edits become successful ones after FT, that the target knowledge involves high-frequency tokens. For instance, when querying the mother tongue of Danielle Darrieux, the expected answer from $M_{ed}$ is English, but the actual output is "United States". After fine-tuning, however, $M_{ed\_ft}$ produces "English", which is a frequent word.

We further observe that once an edit is erased by fine-tuning, the model does not necessarily revert to the original answer but often defaults to a higher-frequency alternative with similar semantics or word class. For example, in the third case of *Erased Edits*, after the target "Philadelphia" is removed, $M_{ed\_ft}$ outputs "London" rather than the original answer "Paris". Further, we find an interesting case that in the final case of *Impossible Edits*, both $M_{ed}$ and $M_{ed\_ft}$ return "1 May 1977", whereas the expected answer is "12 May 1977". This deviation suggests a possible bias from pre-training data related to Labour Day. We leave this to future investigation.

### 4.3 ONLY FINE-TUNING EDITED OR NON-EDITED LAYERS

As discussed in Sec. 2.3, our motivation also lies in understanding how to remove potentially harmful edits and how to preserve beneficial ones to avoid repeated editing. At the same time, as shown above, fine-tuning can remove edits from the edited model. Taken together, we propose two hypotheses: (1) FT edited layers can only effectively remove edits; (2) FT non-edited layers can preserve edits.

To examine this, we set two experimental groups: fine-tuning only the edited layers and fine-tuning only the non-edited layers. For our experiments, we adopt Llama2 and GPT-J as base models, MEMIT and AlphaEdit as KE methods, and LoRA and DoRA as fine-tuning approaches. Specifically, we edit layer $3-8$ for GPT-J and layers $4-8$ for Llama2, following Meng et al. (2023) and Wang et al. (2025).

Table 5: KE performance (%) of Llama2 being edited using AlphaEdit on COUNTERFACT dataset, and then being fine-tuned with selective layers. $M^1$ for model without editing or fine-tuning; $M_{ed}^1$ for edited-only model; $M_{ed\_ft\_all}^3$ for edited-then-finetuned with all layers; $M_{ed\_ft\_edited}^4$ for edited-then-finetuned with edited layers; $M_{ed\_ft\_non-edited}^5$ for edited-then-finetuned with non-edited layers. ES, NS and PS are KE metrics, DS is the average score of downstream tasks.

| KE performance | Llama2 $M^1$ | $M_{ed}^2$ | $M_{ed\_ft\_all}^3$ | $M_{ed\_ft\_edited}^4$ | $M_{ed\_ft\_non-edited}^5$ | $M_{ed}$ | $M_{ed\_ft\_all}$ | $M_{ed\_ft\_edited}$ | $M_{ed\_ft\_non-edited}$ |
|---|---|---|---|---|---|---|---|---|---|
| | | | 100 Edits | | | | 1000 Edits | | |
| ES | 20.00 | 96.00 | 98.00 | 66.00 | 72.00 | 100.00 | 90.55 | 57.80 | 60.30 |
| PS | 35.00 | 87.50 | 93.00 | 68.00 | 64.00 | 95.75 | 77.03 | 60.95 | 54.40 |
| NS | 69.00 | 76.60 | 76.60 | 83.60 | 82.20 | 72.44 | 73.20 | 79.52 | 80.31 |
| DS | 1.77 | 2.15 | 81.7 | 65.43 | 80.61 | 4.79 | 81.00 | 65.35 | 72.46 |

**Fine-tuning only edited layer**    First, we find that fine-tuning only the edited layers can remove more prior edits than fine-tuning all layers. As illustrated in Tab. 5, in the case of 100 Edits, between $M_{ed\_ft\_all}$ and $M_{ed\_ft\_edited}$, $M_{ed\_ft\_edited}$ shows a larger drop of editing performance across all three editing metrics (ES, PS, NS). For example, ES of $M_{ed\_ft\_edited}$ drops to 66% while ES of $M_{ed\_ft\_all}$ rises to 98%, close to 96% of $M_{ed}$. However, fine-tuning only edited layers can result in a loss of downstream performance. For instance, in the case with 1000 edits in Tab. 15, performance on *BoolQ* decreased 3.31% from 71.44% to 68.13%. The only exception is HellaSwag, where performance drops sharply from 89.00% to 32.10%. When jointly considering ES and the overall downstream performance ($\Delta ES$ vs. $\Delta DS$), we observe that although $\Delta ES$ decreases by nearly 45% (from 96% to 66%), the average downstream score of the 100-edit model declines by only 24% (from 81.7% to 65.43%). This underscores the trade-off between effectively removing edits and the risk of losing downstream task performance. If overall downstream performance is not a priority, fine-tuning only the edited layers is an effective strategy for removing unwanted edits.

**Fine-tuning only non-edited layer**    To test whether edits can be preserved by fine-tuning only the non-edited layers, we compare $M_{ed\_ft\_all}$ with $M_{ed\_ft\_non-edited}$ in Tab. 5. The results are negative: **fine-tuning non-edited layers provides no benefit in preserving edits**. For example, with 100 edits using AlphaEdit on Llama2, $M_{ed\_ft\_non-edited}$ shows a significant decline in ES from 98% to 72%, whereas $M_{ed\_ft\_all}$ maintains an ES of 96%, close to its pre-FT value. Evaluations on paraphrased prompts yield similar results, with $M_{ed\_ft\_non-edited}$ exhibiting greater degradation. We further investigate whether fine-tuning only the non-edited layers can effectively remove edits. As shown in Tab. 17, this approach preserves stronger downstream performance (80.61 vs. 65.43; 72.46 vs. 65.35, all in %) but erases fewer edits than fine-tuning only the edited layers. These results suggest that fine-tuning non-edited layers can be a supplementary edit-removal strategy.

**Discussion**    Through above experiments, we have findings as: (i) in edit-removal, $M_{ed\_ft\_edited} > M_{ed\_ft\_non-edited} > M_{ed\_ft\_all}$; (ii) regarding the effectiveness of fine-tuning, we have $M_{ed\_ft\_all} > M_{ed\_ft\_non-edited} > M_{ed\_ft\_edited}$. Above observation aligns with the **distributed representation hypothesis**, which posits that factual associations in LLMs emerge from coordinated patterns across many MLP and attention layers (Geva et al., 2023; Dar et al., 2023). Editing, which modifies only a subset of weights, leads to incomplete shifts within these distributed circuits. Fine-tuning can readily disrupt the coordinated structure that supports the edits.

## 4.4 EDITING PERFORMANCE OF DEEPSEEK

In our editing experiments, both AlphaEdit and MEMIT perform poorly on DeepSeek. We verified this result using editing layers identified via causal tracing as well as the default Llama settings, the backbone of the distilled DeepSeek model used here. Detailed results are provided in App. G. The limited number of successful edits makes it difficult to fairly assess the impact of fine-tuning on editing, rendering further experiments unnecessary. This underscores the lack of robustness in current KE methods and their unsuitability for emerging models such as DeepSeek.

## 4.5 ABLATION ANALYSIS

**KE impact on FT Performance on Downstream**    To assess the impact of editing on subsequent fine-tuning, we compare the downstream performance between $M_{ft}$ and $M_{ed\_ft}$. We find that **KE moderately reduces the effectiveness of subsequent fine-tuning, even when applied before-**

Table 6: Average scores (%) across downstream tasks for groups of all FT methods (Full fine-tuning, LoRA, DoRA) with editing number ranging from 0 to $10^4$, respectively. KE[1] methods include MEMIT[2] and AlphaEdit[3], CF[4] refers to COUNTERFACT dataset. Cyan indicates decline in downstream performance while Orange represents increase in performance.

| Model | Dataset | KE[1] | No fine-tuning | | | | Full fine-tuning | | | | LoRA | | | | DoRA | | | |
|---|---|---|---|---|---|---|---|---|---|---|---|---|---|---|---|---|---|---|
| | | | 0 | $10^2$ | $10^3$ | $10^4$ | 0 | $10^2$ | $10^3$ | $10^4$ | 0 | $10^2$ | $10^3$ | $10^4$ | 0 | $10^2$ | $10^3$ | $10^4$ |
| GPT-J | zsRE | M[2] | 11.56 | 4.64 | 5.77 | 5.72 | 37.41 | 36.95 | 40.92 | 39.69 | 64.24 | 60.8 | 54.67 | 63.98 | 67.87 | 67.05 | 67.72 | 65.08 |
| | | AE[3] | | 4.66 | 4.47 | 3.47 | | 32.51 | 30.58 | 24.73 | | 64.35 | 61.56 | 47.50 | | 67.60 | 60.85 | 60.75 |
| | CF[4] | M | | 3.60 | 6.34 | 10.30 | | 38.50 | 38.64 | 32.65 | | 60.79 | 66.88 | 65.16 | | 64.29 | 60.82 | 61.71 |
| | | AE | | 3.37 | 2.16 | 5.37 | | 37.29 | 33.75 | 30.52 | | 67.09 | 60.24 | 57.88 | | 67.81 | 63.69 | 59.25 |
| Llama2 | zsRE | M | 1.77 | 0.83 | 14.68 | 7.28 | 54.62 | 35.75 | 33.46 | 35.5 | 79.87 | 77.95 | 70.27 | 70.27 | 80.1 | 78.95 | 69.61 | 63.62 |
| | | AE | | 1.57 | 2.01 | 6.16 | | 38.88 | 40.51 | 26.01 | | 80.49 | 79.67 | 78.85 | | 80.58 | 80.02 | 78.85 |
| | CF | M | | 7.65 | 7.02 | 6.77 | | 38.85 | 37.22 | 34.14 | | 72.74 | 71.71 | 63.68 | | 78.66 | 71.51 | 64.05 |
| | | AE | | 1.50 | 3.28 | 6.44 | | 31.53 | 29.75 | 29.54 | | 75.18 | 36.11 | 77.88 | | 80.25 | 79.98 | 78.16 |
| GPT2-XL | zsRE | M | 13.35 | 13.52 | 13.26 | 12.19 | 28.58 | 28.88 | 29.46 | 26.74 | 35.29 | 38.99 | 38.4 | 35.53 | 36.69 | 38.59 | 38.59 | 35.64 |
| | | AE | | 11.74 | 14.93 | 13.13 | | 28.55 | 28.63 | 28.38 | | 27.3 | 33.69 | 34.73 | | 27.37 | 33.68 | 31.31 |
| | CF | M | | 14.68 | 11.80 | 17.72 | | 28.08 | 29.31 | 30.01 | | 30.31 | 34.30 | 33.99 | | 30.28 | 34.37 | 34.79 |
| | | AE | | 12.84 | 16.65 | 5.02 | | 29.17 | 28.33 | 28.71 | | 36.13 | 35.82 | 35.54 | | 30.07 | 34.93 | 31.70 |

**hand**, and the level of impact depends on settings, such as FT methods and KE-related settings. As shown in Tab. 6, most $M_{\text{ed\_ft}}$ exhibit performance degradation on downstream tasks compared to their counterparts $M_{\text{ft}}$, cross KE methods, and editing datasets. Of the 144 fine-tuned cases, 122 cases experience decline, while the remaining cases exhibit an increase in downstream performance. The largest decline occurs when Llama2 is edited via AlphaEdit on 100 counterfactual facts, followed by LoRA fine-tuning, resulting in a drop of 43.76 p.p. in accuracy rate (from 79.87% to 36.11%). Among the three fine-tuning methods, DoRA is the most severely affected by KE, as indicated by the predominance of green cells, whereas full fine-tuning is the most robust, with 11 out of 36 configurations showing even improved performance. A detailed analysis of downstream task performance across experiment settings is provided in App. H.1.

**Catastrophic Forgetting** We further examine whether catastrophic forgetting, rather than KE, drives the observed edit decay. To do so, we compare the edited-then-finetuned model $M_{\text{ed\_ft}}$ with its fine-tuned-only counterpart $M_{\text{ft}}$ on downstream tasks. The results indicate that catastrophic forgetting is unlikely the cause: if it were, both $M_{\text{ed\_ft}}$ and $M_{\text{ft}}$ would show performance drops. Instead, as shown in Tab. 6, $M_{\text{ed\_ft}}$ (77.95) achieves comparable downstream performance to $M_{\text{ft}}$ (1.77), both outperforming $M_{\text{ed}}$ (0.83) and $M$ (0.83). This pattern suggests that $M_{\text{ed\_ft}}$ does not have catastrophic forgetting issue.

## 5 WHY EDITING IS FRAGILE TO FINE-TUNING

Building on Sec. 4.3, which shows that fine-tuning non-edited layers can degrade editing performance, we note that factual associations in LLMs are encoded via distributed mechanisms across multiple layers and directions within the residual stream (Dar et al., 2023; Geva et al., 2023; Choe et al., 2025). Motivated by this, we investigate whether a knowledge edit induces a coherent shift in activation space and how subsequent fine-tuning affects it. For a model $M$, its edited version $M_{\text{ed}}$, and its edited-then-fine-tuned version $M_{\text{ed\_ft}}$, we analyze activations $h_\ell(x)$ at each layer $\ell$ using prompt $x$ from a diagnostic prompt set $\mathcal{X}$, which comprises two groups: prompts that explicitly query the edited knowledge (from the editing dataset, Sec. 3.1) and prompts that do not directly invoke the edited fact (from downstream tasks, Sec. 3.1).

**Layer-wise drift** For each prompt $x$, we compute the magnitude of activation changes introduced by editing ($\Delta_\ell^{\text{ed}}(x)$), by fine-tuning ($\Delta_\ell^{\text{ft}}(x)$) and by both ($\Delta_\ell^{\text{ed\_ft}}(x)$), where :

$$\Delta_\ell^{\text{ed}}(x) = \|h_\ell^{M_{\text{ed}}}(x) - h_\ell(x)\|_2, \; \Delta_\ell^{\text{ft}}(x) = \|h_\ell^{M_{\text{ft}}}(x) - h_\ell(x)\|_2, \; \Delta_\ell^{\text{ed\_ft}}(x) = \|h_\ell^{M_{\text{ed\_ft}}}(x) - h_\ell^{M_{\text{ed}}}(x)\|_2. \quad (2)$$

We compute the *arithmetic mean* over all prompts and visualize activation changes across layers in Fig. 3. We observe that (i) fine-tuned models ($M_{\text{ft}}$, $M_{\text{ed\_ft}}$) exhibit larger activation changes than non-fine-tuned models, and (ii) edited-only models show changes primarily from the edited layer, while fine-tuning affects a broader range of layers. These findings suggest that edits induce shallow, localized activation perturbations, which can be overwritten by the broader effects of fine-tuning, as discussed in Sec. 4.3.

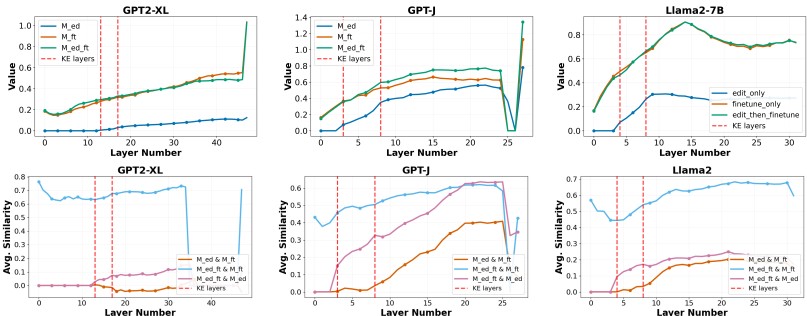

Figure 3: In **layer-wise activation drifts** (upper three) for GPT2-XL, GPT-J and Llama2, 3 categories for each model: $M_{\text{ed}}$, $M_{\text{ft}}$ and $M_{\text{ed\_ft}}$. In **directional similarities (bottom three)**, 3 pairs of categories tested for each model: $M_{\text{ed}}$ - $M_{\text{ft}}$, $M_{\text{ed\_ft}}$ - $M_{\text{ft}}$ and $M_{\text{ed\_ft}}$ - $M_{\text{ed}}$. Within the red vertical dash lines are the range of layers being edited. Result specifications in App. I

**Directional similarity** To further characterise how fine-tuning interacts with the directions introduced by editing, we compute the cosine similarity between the *editing direction* and the *fine-tuning direction* in activation space. We first define the layer-wise displacement vectors using Equ. 4. We then compute the layer-wise directional similarity by averaging similarities for single prompts $x$ ($x \in \mathcal{X}$) at layer $\ell$, as shown in Equ. 4 ($\varepsilon = 10^{-8}$ is used to prevent division by zero):

$$\Delta_{\ell}^{\text{ed}}(x) = h_{\ell}^{M_{\text{ed}}}(x) - h_{\ell}(x), \ \Delta_{\ell}^{\text{ft}}(x) = h_{\ell}^{M_{\text{ft}}}(x) - h_{\ell}(x), \ \Delta_{\ell}^{\text{ed\_ft}}(x) = h_{\ell}^{M_{\text{ed\_ft}}}(x) - h_{\ell}^{M_{\text{ed}}}(x). \quad (3)$$

$$\text{sim}_{\ell}(x) = \frac{1}{|\mathcal{X}|} \sum_{x \in \mathcal{X}} \frac{\langle \Delta_{\ell}^{1}(x), \ \Delta_{\ell}^{2}(x) \rangle}{\|\Delta_{\ell}^{M_1}(x)\|_2 \ \|\Delta_{\ell}^{M_2}(x)\|_2 + \varepsilon}. \quad (4)$$

A value $\text{sim}_{\ell} \approx 1$ would indicate that fine-tuning pushes activations further in the direction of the KE, whereas a negative value suggests they are in the opposite direction. As shown in the bottom row of Fig. 3, fine-tuned models ($M_{\text{ed\_ft}}$ - $M_{\text{ft}}$) shares the lowest similarity, indicating that fine-tuning moves activations in the positive directions *nearly orthogonal to* the editing direction. This orthogonality helps explain why edits are overwritten even when kept in their original layers.

**Discussion** The results show that $M_{\text{ft}}$ and $M_{\text{ed\_ft}}$ exhibit both the largest activation-magnitude changes and the highest activation similarity, indicating that fine-tuning, not KE, overwhelmingly dominates models' representations. In contrast, $M_{\text{ed}}$ has small, dispersed activation shifts which are deviant to the directions introduced by fine-tuning. The above findings indicates that edited knowledge may be overwritten by fine-tuned knowledge during representation.

## 6 CONCLUSION

In this paper, we show that knowledge edits rarely persist unchanged under fine-tuning: in many cases, fine-tuning impairs editing performance or even elicits new knowledge that is different from the target and original knowledge. At the same time, we find that edits themselves can affect downstream fine-tuning performance, even when applied. Motivated by the dual goals of preserving beneficial edits and removing malicious ones, we further explored selective fine-tuning strategies. The results show that updating only non-edited layers to preserve beneficial edits slightly sacrifices downstream performance. For removing covert edits, tuning edited layers does not help and calls for future exploration. These results establish that model editing and fine-tuning are tightly coupled processes whose interaction can be exploited to balance adaptability with knowledge control. Our work provides both empirical baselines and actionable strategies for building large language models that remain adaptable yet reliably steerable with respect to edited knowledge. Future research on model editing should consider robustness not in isolation but across the entire LLM pipeline.

## 7 REPRODUCIBILITY STATEMENT

We provide all necessary resources to facilitate reproducibility of our results. Dataset descriptions and preprocessing steps are detailed in Sec. 3.1 and App. K. Implementation details, model configurations, and training setups are reported in App. B and App. K. We will release the code to reproduce all experiments once published. Together, these materials ensure that our results can be independently verified.

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

## A  IMPLEMENTATION DETAILS

We build on the `MEMIT` codebase[5] and EasyEdit[6], and implement all fine-tuning with Hugging Face Transformers (v4.43) in PyTorch, using A100 GPUs with `bf16` precision. Models include `GPT-J-6B`, `GPT-2XL`, and `Llama-2-7B`. Knowledge edits are applied with MEMIT and AlphaEdit following their default setups. Fine-tuning uses three methods: Full FT, LoRA (r=8, $\alpha = 16$, dropout=0.05), and DoRA. Edited vs. non-edited layer experiments freeze parameters outside the specified layers. Optimizers are AdamW with learning rate $1e-5$–$5e-5$, batch size 64, and 2–3 epochs.

## B  MODEL CONFIGURATIONS

Each column in Tab. 8 represents a selectable parameter, with experimental settings generated by their Cartesian product. As noted in its caption, configurations with zero edits are consistent across datasets, e.g., Llama2 edited with 0 facts from zsRE and COUNTERFACT dataset are identical. A summary of model details is provided in Tab. 7. Additionally, DeepSeek is not fine-tuned, with the rationale detailed in Sec. 4.4.

Table 7: Summary of model details.

| Model name | Year of release | Num of parameters | Huggingface handle |
|---|---|---|---|
| GPT2-XL | 2019 | 1.61B | openai-community/gpt2-xl |
| GPT-J-6B | 2021 | 6.05B | EleutherAI/gpt-j-6b |
| Llama2-7B-hf | 2023 | 6.74B | meta-llama/Llama-2-7b-hf |
| Llama3.1-8B-Instruct | 2024 | 8.03B | meta-llama/Llama-3.1-8B-Instruct |
| DeepSeek-R1-Distill-Llama-8B | 2025 | 8.03B | DeepSeek-R1-Distill-Llama-8B |

Table 8: Scope of the experimental parameters. M - models, D - datasets, E - editing methods, N - editing numbers, F - fine-tuning methods. Note that settings with 0 edit are identical across datasets, e.g., GPT-J with 0 edits on zsRE is identical to GPT-J with 0 edits on COUNTERFACT.

| Model | | Dataset | | Edit method | | Edits | | fine-tune method |
|---|---|---|---|---|---|---|---|---|
| GPT2-XL | | | | | | 0 | | No fine-tuning |
| GPTJ | | zsRE | | No editing | | 100 | | LoRA |
| Llama2 | $\times$ | COUNTERFACT | $\times$ | MEMIT | $\times$ | 1000 | $\times$ | DoRA |
| Llama3.1 | | | | AlphaEdit | | 10000 | | Full-size |
| DeepSeek | | | | | | | | |

$$S = M \times D \times E \times N \times F$$
$$= \{(m, d, e, n, f) \mid m \in M, d \in D, e \in E, n \in N, f \in F\}$$

## C  RESULTS VALIDATION

In this section, "original values" refers the results given by the paper and "validating values" refers to the results obtained in the validation experiments. Overall, our validation demonstrates that the KE and fine-tuning results produced by our code are highly consistent with the original results, indicating that the outputs generated by our implementation are relatively reliable. Minor discrepancies may arise from factors such as model loading precision, random initialization, or hardware-related numerical differences.

---

[5] https://github.com/kmeng01/memit
[6] https://github.com/zjunlp/EasyEdit

Table 9: KE performances (%) check between original values put forward in paper and our validating results. Comparison across various KE method (MEMIT, AlphaEdit) and datasets (zsRE and COUNTERFACT).

| | MEMIT | | | | AlphaEdit | | | |
| Metrics | zsRE | | COUNTERFACT | | zsRE | | COUNTERFACT | |
| | Original values | Validating values | Original values | Validating values | Original values | Validating values | Original values | Validating values |
|---|---|---|---|---|---|---|---|---|
| Efficacy | 96.70(±0.30) | 96.93 | 98.9 (±0.20) | 99.10 | 99.79 (±0.14) | 99.31 | 99.75 (±0.08) | 98.35 |
| Paraphrase | 89.70(±0.50) | 90.75 | 88.6 (±0.50) | 88.66 | 96.00 (±0.22) | 96.71 | 96.38 (±0.23) | 95.90 |
| Specificity | 26.60 (±0.30) | 26.33 | 73.70 (±0.50) | 73.53 | 28.29 (±0.25) | 28.07 | 75.48 (±0.21) | 80.16 |

**Validation for KE**   As shown in Tab. 9, the results obtained from our validation experiments closely align with the original values reported in the paper, indicating strong reproducibility and correctness of our re-implementation.

**Validation for fine-tuning**   As shown in Fig. 4 and Fig. 5, validating results are close to original results, indicating the reliability of our outputs.

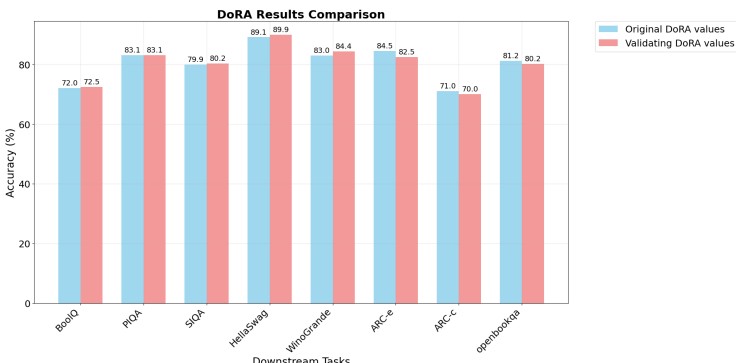

Figure 4: Difference between validating values and original values across eight downstream tasks.

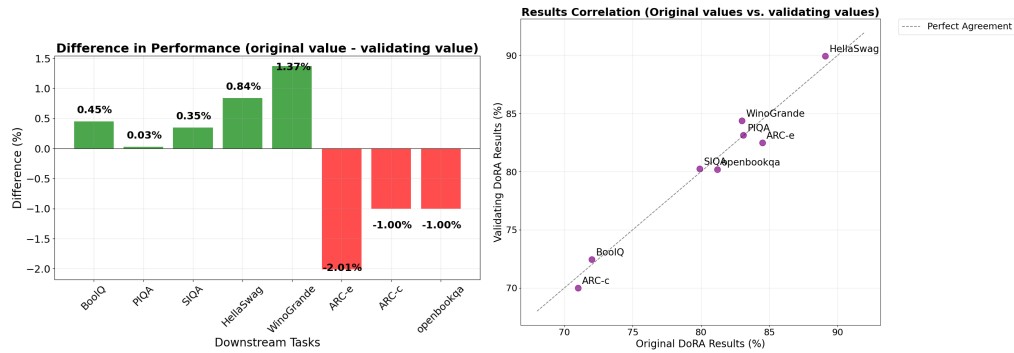

Figure 5: Difference between validating values and original values in ratio across eight downstream tasks.

## D   KNOWLEDGE EDITING METRICS

We construct our evaluation using metrics defined in previous works(Meng et al., 2022; Yao et al., 2023). For each edit instance $i$ from zsRE in an edited model $M_{ed}$, we set $s_i$ to be the subject, $r_i$ to be the relation, and $o_i$ to be the target object. We write $p(s_i, r_i)$ for the base prompt constructed from $(s_i, r_i)$. For COUNTERFACT, we additionally denote by $o_i^c$ the original (counterpart) object describing the real-world fact, and by paraphrases$(s_i, r_i)$ and neighborhood$(s_i, r_i)$ the sets of paraphrased and neighborhood prompts, respectively. Given a prompt $p$, we use $Pr_{M_{ed}}(x \mid p)$ for the model's predicted probability of token $x$.

**ES Success (Efficacy / ES).** Efficacy measures the proportion of successful edits. Formulas 5 and 6 determine the successful editing of an editing instance from zsRE and COUNTERFACT datasets, respectively. For zsRE, an edit is considered successful on instance $i$ if the edited model assigns the highest probability to the desired answer $o_i$ under the base prompt $p(s_i, r_i)$ (Equ. 5). For COUNTERFACT, each edit $i$ specifies a counterfactual object $o_i$ to be written and a corresponding real-world object $o_i^c$. An edit $i$ is considered successful, under the base prompt $p(s_i, r_i)$, if the edited model assigns higher probability to the desired counterfactual $o_i$ than to the original object $o_i^c$ (Equ. 6):

$$\text{ES}_i^{\text{zsRE}} = \mathbf{1}\Big(o_i = \arg\max_x Pr_{M_{\text{ed}}}(x \mid p(s_i, r_i))\Big) \tag{5}$$

$$\text{ES}_i^{\text{CF}} = \mathbf{1}\Big(Pr_{M_{\text{ed}}}(o_i \mid p(s_i, r_i)) > Pr_{M_{\text{ed}}}(o_i^c \mid p(s_i, r_i))\Big) \tag{6}$$

Thus, the overall *ES* can be calculated as:

$$\text{ES}^{\text{CF/zsRE}} = \frac{1}{N} \sum_{i=1}^{N} \text{ES}_i^{\text{CF/zsRE}} \tag{7}$$

**Paraphrase Success (Paraphrase / PS).** Paraphrase evaluates model's generalization ability after editing facts. We consider for each instance $i$ a set of paraphrases $paraphrases(s_i, r_i)$ of the base prompt. For zsRE, we evaluate average top-1 accuracy on rephrased prompts $N(s_i, r_i)$. For COUNTERFACT, On a rephrased prompt $p \in paraphrases(s_i, r_i)$, we declare success if the model again prefers the counterfactual object over the original. Formulas 8 and 9 present the mathematical definition of Paraphrase for the zsRE and COUNTERFACT datasets, respectively:

$$\text{PS}^{\text{zsRE}} = \frac{1}{N} \sum_{i=1}^{N} \mathbf{1}\Big(o_i = \arg\max_o Pr_{M_{\text{ed}}}(o \mid N(s_i, r_i))\Big) \tag{8}$$

$$\text{PS}_i^{\text{CF}} = \frac{1}{N} \sum_{i=1}^{N} \mathbf{1}\Big(Pr_{M_{\text{ed}}}(o_i \mid p) > Pr_{M_{\text{ed}}}(o_i^c \mid p)\Big) \tag{9}$$

**Neighborhood Success (Specificity / NS).** Specificity assesses the locality of a knowledge edit by measuring its unwanted impact on facts unrelated to the facts involved in KE. To obtain NS, we consider for each instance $i$ a set of neighborhood prompts $neighborhood(s_i, r_i)$ that should *not* be affected by the edit. Formulas 10 and 11 present the mathematical definition of Specificity for the zsRE and COUNTERFACT datasets, respectively. For zsRE, $O(s_i, r_i)$ represents the unrelated facts:

$$\text{NS}^{\text{zsRE}} = \frac{1}{N} \sum_{i=1}^{N} \mathbf{1}\Big(o_i = \arg\max_o Pr_{M_{\text{ed}}}(o \mid O(s_i, r_i))\Big) \tag{10}$$

$$\text{NS}^{\text{CF}} = \frac{1}{N} \sum_{i=1}^{N} \mathbf{1}\Big(Pr_{M_{\text{ed}}}(o_i \mid p) < Pr_{M_{\text{ed}}}(o_i^c \mid p)\Big) \tag{11}$$

# E  OVERALL KE PERFORMANCE

**MEMIT and AlphaEdit**  The full set of experimental combinations mentioned in below charts can be found in Tab. 8. Tab. 10 presents the overall knowledge editing (KE) results for GPT-J across different combinations of KE dataset, number of edits, KE method, and fine-tuning method. Tab. 11 reports the corresponding results for Llama2 under the same experimental configurations while Tab. 12 shows the results for GPT2-XL. Tab. 13 presents the KE results for Llama3.1.

Table 10: GPT-J's KE performance (%) under different FT settings (No fine-tuning[2], LoRA, DoRA and full fine-tuning[3]) across various KE method (MEMIT, AlphaEdit) and datasets (zsRE and COUNTERFACT[4]).

| Dataset | #Edits | Metrics | MEMIT | | | | AlphaEdit | | | |
|---|---|---|---|---|---|---|---|---|---|---|
| | | | No ft | LoRA | DoRA | Full ft | No ft[2] | LoRA | DoRA | Full ft[3] |
| zsRE | 0 | ES | 23.47 | 23.42 | 22.50 | 22.67 | 23.47 | 23.42 | 22.50 | 22.67 |
| | | PS | 23.17 | 21.56 | 21.65 | 23.50 | 23.17 | 21.56 | 21.65 | 23.50 |
| | | NS | 28.35 | 27.82 | 27.76 | 25.93 | 28.35 | 27.82 | 27.76 | 25.93 |
| | $10^2$ | ES | 99.07 | 88.79 | 89.87 | 97.95 | 99.33 | 84.74 | 99.33 | 98.50 |
| | | PS | 95.76 | 81.09 | 84.62 | 95.15 | 97.55 | 76.00 | 97.55 | 96.93 |
| | | NS | 28.74 | 27.12 | 27.64 | 27.25 | 28.65 | 26.12 | 28.65 | 27.57 |
| | $10^3$ | ES | 99.10 | 84.53 | 85.31 | 98.74 | 99.31 | 84.74 | 81.70 | 98.98 |
| | | PS | 95.97 | 77.77 | 78.48 | 94.05 | 96.71 | 76.00 | 73.82 | 94.39 |
| | | NS | 28.13 | 26.80 | 25.87 | 27.06 | 28.07 | 26.12 | 26.25 | 26.12 |
| | $10^4$ | ES | 96.93 | 66.52 | 67.83 | 89.38 | 89.81 | 49.97 | 23.64 | 74.62 |
| | | PS | 90.75 | 58.92 | 61.22 | 81.21 | 78.59 | 43.90 | 23.03 | 64.57 |
| | | NS | 26.33 | 25.42 | 24.50 | 25.54 | 22.96 | 23.89 | 25.57 | 22.96 |
| CF[4] | 0 | ES | 15.00 | 15.00 | 16.00 | 14.00 | 15.00 | 15.00 | 16.00 | 14.00 |
| | | PS | 16.50 | 23.50 | 20.00 | 18.50 | 16.50 | 23.50 | 20.00 | 18.50 |
| | | NS | 84.40 | 82.60 | 83.50 | 84.80 | 84.40 | 82.60 | 83.50 | 84.80 |
| | $10^2$ | ES | 100.00 | 100.00 | 100.00 | 100.00 | 100.00 | 99.00 | 100.00 | 100.00 |
| | | PS | 95.00 | 92.50 | 91.00 | 91.90 | 98.50 | 94.00 | 94.00 | 97.00 |
| | | NS | 81.10 | 80.90 | 81.80 | 81.62 | 81.10 | 80.00 | 80.50 | 81.40 |
| | $10^3$ | ES | 100.00 | 99.00 | 99.00 | 99.90 | 98.35 | 99.70 | 99.60 | 97.95 |
| | | PS | 93.95 | 87.70 | 86.90 | 90.45 | 95.90 | 90.90 | 90.95 | 97.00 |
| | | NS | 81.17 | 79.96 | 80.80 | 80.68 | 80.16 | 79.58 | 79.84 | 81.40 |
| | $10^4$ | ES | 99.10 | 94.34 | 94.46 | 97.79 | 98.87 | 99.16 | 82.00 | 95.85 |
| | | PS | 88.66 | 76.27 | 76.24 | 80.38 | 86.70 | 87.62 | 62.00 | 74.09 |
| | | NS | 73.53 | 73.89 | 74.17 | 75.08 | 67.77 | 68.54 | 74.00 | 72.56 |

Table 11: Llama2's KE performance (%) under different FT settings (No fine-tuning[2], LoRA, DoRA and full fine-tuning[3]) across various KE method (MEMIT, AlphaEdit) and datasets (zsRE and COUNTERFACT[4]).

| Dataset | #Edits | Metrics | MEMIT | | | | AlphaEdit | | | |
|---|---|---|---|---|---|---|---|---|---|---|
| | | | No ft | LoRA | DoRA | Full ft | No ft[2] | LoRA | DoRA | Full ft[3] |
| zsRE | 0 | ES | 45.61 | 46.92 | 42.30 | 22.67 | 45.61 | 46.92 | 42.30 | 22.67 |
| | | PS | 45.57 | 43.08 | 42.02 | 23.5 | 45.57 | 43.08 | 42.02 | 23.50 |
| | | NS | 32.15 | 36.85 | 28.25 | 25.93 | 32.15 | 36.85 | 28.25 | 25.93 |
| | $10^2$ | ES | 86.03 | 76.30 | 72.00 | 22.67 | 93.33 | 57.96 | 93.33 | 21.83 |
| | | PS | 86.01 | 71.69 | 66.02 | 20.14 | 84.93 | 53.49 | 84.93 | 19.84 |
| | | NS | 31.68 | 29.69 | 28.70 | 14.58 | 32.42 | 29.67 | 32.42 | 15.34 |
| | $10^3$ | ES | 51.38 | 46.52 | 46.00 | 10.22 | 93.23 | 50.45 | 51.53 | 24.36 |
| | | PS | 50.04 | 44.44 | 46.65 | 10.37 | 86.57 | 48.79 | 48.83 | 22.27 |
| | | NS | 28.09 | 27.68 | 25.79 | 12.22 | 34.3 | 32.11 | 30.88 | 15.40 |
| | $10^4$ | ES | 48.62 | 48.64 | 48.23 | 14.00 | 84.31 | 46.03 | 45.38 | 24.44 |
| | | PS | 50.20 | 49.75 | 48.62 | 13.20 | 79.03 | 44.27 | 43.65 | 21.97 |
| | | NS | 25.73 | 24.97 | 24.32 | 14.59 | 34.35 | 30.45 | 31.98 | 15.59 |
| CF[4] | 0 | ES | 11.37 | 11.41 | 11.52 | 45.00 | 11.37 | 11.41 | 11.52 | 45.00 |
| | | PS | 41.61 | 40.57 | 40.32 | 32.00 | 41.61 | 40.57 | 40.32 | 32.00 |
| | | NS | 91.36 | 91.32 | 91.21 | 50.00 | 91.36 | 91.32 | 91.21 | 50.00 |
| | $10^2$ | ES | 100.00 | 94.00 | 97.00 | 47.00 | 100.00 | 66.00 | 61.00 | 48.00 |
| | | PS | 98.00 | 81.50 | 95.50 | 70.50 | 77.50 | 49.00 | 49.50 | 59.50 |
| | | NS | 75.40 | 78.50 | 80.60 | 52.00 | 85.50 | 84.30 | 83.80 | 52.30 |
| | $10^3$ | ES | 100.00 | 94.00 | 94.50 | 47.60 | 99.10 | 52.40 | 56.90 | 48.60 |
| | | PS | 94.50 | 82.50 | 84.30 | 67.55 | 67.80 | 42.45 | 42.85 | 54.35 |
| | | NS | 70.50 | 77.00 | 76.33 | 53.02 | 83.97 | 81.22 | 82.20 | 52.06 |
| | $10^4$ | ES | 86.96 | 70.18 | 68.27 | 48.07 | 87.43 | 37.90 | 37.15 | 48.46 |
| | | PS | 73.62 | 64.32 | 61.25 | 32.94 | 55.71 | 34.01 | 33.40 | 47.74 |
| | | NS | 68.64 | 62.35 | 61.74 | 52.71 | 80.67 | 79.02 | 79.89 | 52.52 |

Table 12: GPT2-XL's KE performance (%) under different FT settings (No fine-tuning[2], LoRA, DoRA and full fine-tuning[3]) across various KE method (MEMIT, AlphaEdit) and datasets (zsRE and COUNTERFACT[4]).

| Dataset | #Edit | Metrics | MEMIT | | | | AlphaEdit | | | |
|---|---|---|---|---|---|---|---|---|---|---|
| | | | No ft[2] | LoRA | DoRA | Full ft | No ft | LoRA | DoRA | Full ft[3] |
| zsRE | 0 | ES | 32.80 | 32.74 | 32.87 | 18.04 | 32.80 | 32.74 | 32.87 | 18.04 |
| | | PS | 35.60 | 35.32 | 35.62 | 17.57 | 35.60 | 35.32 | 35.62 | 17.57 |
| | | NS | 23.76 | 23.70 | 23.75 | 24.64 | 23.76 | 23.70 | 23.75 | 24.64 |
| | $10^2$ | ES | 80.00 | 58.37 | 58.39 | 81.16 | 97.18 | 67.80 | 76.72 | 18.04 |
| | | PS | 76.10 | 52.94 | 53.86 | 74.30 | 93.60 | 58.42 | 64.35 | 17.57 |
| | | NS | 25.75 | 24.57 | 24.98 | 25.79 | 25.06 | 27.11 | 24.48 | 24.64 |
| | $10^3$ | ES | 77.85 | 43.51 | 45.18 | 81.22 | 93.13 | 54.52 | 55.11 | 24.74 |
| | | PS | 73.42 | 42.87 | 43.62 | 74.92 | 87.03 | 48.61 | 50.64 | 23.86 |
| | | NS | 26.22 | 23.57 | 24.35 | 25.76 | 25.22 | 25.43 | 26.07 | 24.00 |
| | $10^4$ | ES | 62.61 | 20.34 | 20.65 | 63.39 | 62.34 | 25.80 | 27.47 | 22.09 |
| | | PS | 57.68 | 19.63 | 19.89 | 57.87 | 54.84 | 24.36 | 26.17 | 21.13 |
| | | NS | 25.81 | 24.59 | 24.76 | 24.83 | 21.31 | 23.69 | 24.17 | 23.67 |
| CF[4] | 0 | ES | 20.00 | 20.13 | 20.14 | 19.00 | 20.00 | 20.13 | 20.14 | 19.00 |
| | | PS | 35.00 | 35.21 | 35.17 | 22.50 | 35.00 | 35.21 | 35.17 | 22.50 |
| | | NS | 69.00 | 68.93 | 68.97 | 79.40 | 69.00 | 68.93 | 68.97 | 79.40 |
| | $10^2$ | ES | 97.00 | 83.00 | 82.00 | 97.00 | 100.00 | 96.00 | 98.00 | 19.00 |
| | | PS | 86.50 | 71.00 | 70.50 | 84.50 | 98.00 | 87.50 | 93.00 | 22.00 |
| | | NS | 76.40 | 77.30 | 77.50 | 76.20 | 73.90 | 76.60 | 76.60 | 79.40 |
| | $10^3$ | ES | 93.40 | 78.37 | 78.50 | 92.60 | 100.00 | 89.30 | 90.55 | 21.92 |
| | | PS | 81.35 | 64.43 | 63.45 | 79.55 | 95.75 | 74.55 | 77.03 | 24.66 |
| | | NS | 75.32 | 76.27 | 76.45 | 75.62 | 72.44 | 75.25 | 73.20 | 78.20 |
| | $10^4$ | ES | 79.17 | 62.10 | 61.97 | 78.03 | 92.94 | 53.91 | 57.65 | 21.92 |
| | | PS | 65.44 | 44.47 | 44.43 | 63.74 | 76.33 | 41.94 | 45.77 | 24.66 |
| | | NS | 69.83 | 63.37 | 63.53 | 70.16 | 64.68 | 71.81 | 70.32 | 78.20 |

Table 13: Llama3.1's KE performance (%) under different FT settings (No fine-tuning[2], LoRA, DoRA and full fine-tuning[3]) across various KE method (MEMIT, AlphaEdit) and datasets (zsRE and COUNTERFACT[4]).

| Dataset | #Edits | Metrics | MEMIT | | | | AlphaEdit | | | |
|---|---|---|---|---|---|---|---|---|---|---|
| | | | No ft | LoRA | DoRA | Full ft | No ft[2] | LoRA | DoRA | Full ft[3] |
| zsRE | 0 | ES | 51.10 | 44.92 | 49.01 | 36.55 | 57.62 | 50.77 | 55.12 | 43.88 |
| | | PS | 72.69 | 65.12 | 59.43 | 62.01 | 52.07 | 47.12 | 49.55 | 39.77 |
| | | NS | 37.62 | 31.12 | 29.43 | 23.01 | 69.02 | 62.12 | 64.55 | 54.77 |
| | $10^2$ | ES | 55.32 | 49.12 | 41.43 | 47.01 | 98.24 | 98.32 | 83.55 | 85.77 |
| | | PS | 73.32 | 66.12 | 58.43 | 61.01 | 93.67 | 87.12 | 79.55 | 81.77 |
| | | NS | 57.44 | 51.12 | 43.43 | 45.01 | 47.44 | 41.12 | 33.55 | 35.77 |
| | $10^3$ | ES | 57.04 | 50.12 | 42.43 | 45.01 | 96.86 | 89.12 | 81.55 | 83.77 |
| | | PS | 62.56 | 55.12 | 47.43 | 49.01 | 92.34 | 85.12 | 77.55 | 79.77 |
| | | NS | 32.57 | 26.12 | 18.43 | 21.01 | 48.94 | 42.12 | 34.55 | 36.77 |
| | $10^4$ | ES | 34.92 | 28.12 | 20.43 | 23.01 | 94.43 | 87.12 | 79.55 | 81.77 |
| | | PS | 37.61 | 31.12 | 23.43 | 25.01 | 88.48 | 82.12 | 74.55 | 76.77 |
| | | NS | 16.49 | 10.12 | 16.51 | 15.01 | 36.33 | 30.12 | 22.55 | 24.77 |
| CF[4] | 0 | ES | 7.62 | 5.92 | -2.31 | 3.77 | 12.27 | 9.12 | 12.55 | 7.77 |
| | | PS | 53.61 | 47.12 | 39.43 | 41.01 | 52.03 | 45.12 | 37.55 | 39.77 |
| | | NS | 82.44 | 75.12 | 67.43 | 69.01 | 84.92 | 77.12 | 69.55 | 71.77 |
| | $10^2$ | ES | 99.64 | 92.12 | 84.43 | 87.01 | 99.38 | 92.12 | 84.55 | 86.77 |
| | | PS | 74.03 | 67.12 | 59.43 | 61.01 | 78.63 | 71.12 | 63.55 | 65.77 |
| | | NS | 67.34 | 60.12 | 52.43 | 54.01 | 78.92 | 71.12 | 63.55 | 65.77 |
| | $10^3$ | ES | 98.93 | 91.12 | 83.43 | 85.01 | 99.25 | 92.12 | 84.55 | 86.77 |
| | | PS | 66.56 | 59.12 | 51.43 | 53.01 | 75.25 | 68.12 | 60.55 | 62.77 |
| | | NS | 62.76 | 55.12 | 57.43 | 49.01 | 74.61 | 67.12 | 59.55 | 61.77 |
| | $10^4$ | ES | 69.63 | 62.12 | 54.43 | 56.01 | 98.47 | 91.12 | 83.55 | 85.77 |
| | | PS | 74.82 | 67.12 | 59.43 | 61.01 | 67.87 | 60.12 | 52.55 | 54.77 |
| | | NS | 52.50 | 45.12 | 37.43 | 39.01 | 65.32 | 58.12 | 50.55 | 52.77 |

**MEND** Previous studies have shown that MEND performs poorly on the zsRE dataset, indicating that it is unsuitable for evaluating decay resulting from fine-tuning. Therefore, we did not conduct extensive experiments and instead performed targeted sampling tasks. As shown in Tab. 14, models edited using MEND exhibit patterns consistent with those reported in Sec. 4.1. Specifically, as the number of edits increases, fine-tuning is able to remove a larger proportion of the applied edits. However, MEND performs badly on zsRE dataset. Several studies have demonstrated the the reason: MEND modifies existing weights base on training data, generally performs poorly in such zsro-shot-wise tasks(Fang et al., 2025; Wu et al., 2024).

Table 14: KE performance (%) using MEND under different FT settings (No fine-tuning[2] and DoRA) across datasets (zsRE and COUNTERFACT) and models (GPT2-XL, GPT-J, Llama2, Llama3.1)

| Model | #Edits | zsRE | | COUNTERFACT | |
|---|---|---|---|---|---|
| | | No ft | DoRA | No ft | DoRA |
| GPT2-XL | $10^3$ | 66.57 | 48.92 | 0.00 | 0.00 |
| | $10^4$ | 52.70 | 34.88 | 0.00 | 0.00 |
| GPT-J | $10^3$ | 68.32 | 49.77 | 0.33 | 0.00 |
| | $10^4$ | 45.27 | 27.55 | 0.64 | 0.00 |
| Llama2 | $10^3$ | 71.15 | 53.02 | 0.52 | 0.00 |
| | $10^4$ | 53.24 | 35.88 | 0.31 | 0.00 |
| Llama3.1 | $10^3$ | 82.15 | 63.44 | 0.73 | 0.00 |
| | $10^4$ | 62.17 | 44.55 | 0.57 | 0.00 |

## F   ONLY FINE-TUNING EDITED OR NON-EDITED LAYERS

For this secion, we choose two models for our experiments: Llama2 and GPT-J. Llama2 is discussed in Sec. 4.3. Tab. 15 shows the detailed downstream performance breakdown. We notice that, for task-specific performance aspect, fine-tuning only the edited layers may substantially degrade critical capabilities (e.g., HellaSwag).

Table 15: Downstream performance (%) of Llama2 being edited using AlphaEdit on COUNTER-FACT dataset, and then DoRA fine-tuned with specific layers. Group settings and naming format are identical to Table 5.

| Downstream tasks | Llama2 $M$ | $M_{ed}$ | $M_{ed\_ft\_all}$ | 100 Edits $M_{ed\_ft\_edited}$ | $M_{ed\_ft\_non-edited}$ | $M_{ed}$ | $M_{ed\_ft\_all}$ | 1000 Edits $M_{ed\_ft\_edited}$ | $M_{ed\_ft\_non-edited}$ |
|---|---|---|---|---|---|---|---|---|---|
| BoolQ | 10.31 | 7.68 | 72.14 | 69.27 | 71.04 | 20.92 | 71.44 | 68.13 | 59.51 |
| PIQA | 0.16 | 0.16 | 83.46 | 77.75 | 82.97 | 0.11 | 82.86 | 74.81 | 72.69 |
| SIQA | 2.15 | 2.81 | 80.4 | 75.38 | 79.32 | 2.92 | 79.79 | 76.20 | 69.60 |
| HellaSwag | 0.00 | 0.00 | 89.77 | 29.24 | 88.11 | 0.00 | 89.00 | 32.10 | 80.57 |
| WinoGrande | 0.00 | 0.08 | 82.72 | 75.53 | 81.61 | 0.00 | 81.93 | 75.53 | 79.95 |
| ARC-e, | 0.67 | 0.72 | 83.54 | 79.88 | 82.87 | 0.80 | 83.33 | 79.50 | 79.80 |
| ARC-c | 0.43 | 0.34 | 68.77 | 62.80 | 67.24 | 0.85 | 68.52 | 62.46 | 64.08 |
| openbookqa | 0.40 | 0.20 | 81.20 | 74.80 | 80.80 | 0.60 | 83.00 | 76.60 | 78.00 |
| **Average** | **1.77** | **2.15** | **81.70** | **65.43** | **80.61** | **4.79** | **81.00** | **65.35** | **72.46** |

For **GPT-J**, we choose cases as: (1) GPT-J being edited 100 facts from COUNTERFACT by MEMIT and then fine-tuned by DoRA; (2) GPT-J being edited 100 facts from zsRE by MEMIT and then fine-tuned by DoRA. The result for KE performance is shown in Tab. 16 and the result for downstream performance is shown in Tab. 17.

Table 16: KE performance GPT-J-based model. Naming format are identical to Table 5. Example 1[1]: GPT-J bing edited 100 zsRE facts using MEMIT and fine-tuned by DoRA; Example 2[2]: GPT-J bing edited 100 COUNTERFACT facts using AlphaEdit and fine-tuned by DoRA. GPT-J[3, 4] are GPT-Js without KE and fine-tuning, being evaluated by zsRE and COUNTERFACT, respectively. Naming format are identical to Table 5.

| KE performance | GPT-J[3] $M$ | $M_{ed}$ | $M_{ed\_ft\_all}$ | Example 1[1] $M_{ed\_ft\_edited}$ | $M_{ed\_ft\_non-edited}$ | GPT-J[4] $M$ | $M_{ed}$ | $M_{ed\_ft\_all}$ | Example 2[2] $M_{ed\_ft\_edited}$ | $M_{ed\_ft\_non-edited}$ |
|---|---|---|---|---|---|---|---|---|---|---|
| Efficacy | 23.47 | 99.07 | 89.87 | 95.79 | 86.39 | 15.00 | 100.00 | 100.00 | 100.00 | 100.00 |
| Paraphrase | 23.17 | 95.76 | 84.62 | 91.12 | 84.43 | 16.50 | 98.50 | 94.00 | 97.00 | 94.50 |
| Specificity | 28.35 | 28.74 | 27.64 | 29.13 | 25.71 | 84.40 | 81.10 | 80.50 | 81.00 | 80.40 |

Table 17: Downstream task performance on GPT-J-based examples. Examples choosen are identical to Table 5

| Downstream tasks | GPT-J $M$ | $M_{ed}$ | $M_{ed\_ft\_all}$ | $M_{ed\_ft\_edited}$ Example 1 | $M_{ed\_ft\_non-edited}$ | $M_{ed}$ | $M_{ed\_ft\_all}$ | $M_{ed\_ft\_edited}$ Example 2 | $M_{ed\_ft\_non-edited}$ |
|---|---|---|---|---|---|---|---|---|---|
| BoolQ | 56.57 | 31.04 | 63.88 | 24.95 | 64.40 | 19.27 | 63.67 | 61.99 | 27.92 |
| PIQA | 1.36 | 0.87 | 73.07 | 53.26 | 74.97 | 1.47 | 74.65 | 42.60 | 74.59 |
| SIQA | 0.41 | 0.46 | 73.54 | 63.31 | 74.82 | 0.67 | 74.21 | 41.91 | 69.29 |
| HellaSwag | 0.03 | 0.01 | 70.84 | 42.76 | 59.80 | 0.03 | 71.95 | 18.86 | 33.71 |
| WinoGrande | 30.23 | 0.32 | 69.69 | 59.27 | 67.01 | 0.39 | 70.80 | 37.81 | 69.22 |
| ARC-e, | 1.64 | 1.73 | 67.93 | 21.89 | 67.63 | 1.94 | 68.86 | 56.40 | 65.45 |
| ARC-c | 1.02 | 1.28 | 53.07 | 17.49 | 51.54 | 1.37 | 51.96 | 38.31 | 49.83 |
| openbookqa | 1.20 | 1.40 | 64.40 | 41.40 | 66.00 | 1.80 | 66.40 | 57.60 | 63.80 |
| **Average** | **12.12** | **2.15** | **81.70** | **65.43** | **80.61** | **4.79** | **81.00** | **65.35** | **72.46** |

# G EDITING PERFORMANCE OF DEEPSEEK

## G.1 LAYER DETERMINATION

As there is currently no published research using DeepSeek as base models for KE, we include three potential settings of editing layers when running KE on DeepSeek: (1) using Causal Tracing with Frozen Components (CTFC) to determine layers, (2) directly using LLaMA2's editing layer setting, and (3) directly using GPT2-XL's editing layer setting. The CTFC method is introduced and used by Meng et al. (2023), which enables precise identification of layers most relevant to knowledge storage. Besides, recent KE studies frequently use GPT2-XL and LLaMA2 as base models, directly adopting their layers setting provides reasonable baselines and allows us to carry out comparisons among DeepSeek and them.

## G.2 RESULTS

We ultimately tried all three setups, and the KE results on zsRE dataset with editing number from 0 to 10,000 are shown in Tab. 18. For CTFC, as shown in Fig. 6, layers 1 to 5 shares the largest gap between purple bar and green bar, exhibit the largest gap between the purple and green bars, indicating that these layers contribute most significantly to knowledge storage. Thus, editing layers determined by CTFC are layers 1 to 5. In addition to Fig. 6, we use heatmap (Fig. 7) visualizations of layer-wise causal effects to analyze how different components of the model contribute to factual knowledge retrieval. These heatmaps guide the selection of editing layers by highlighting consistent and concentrated MLP-specific causal effects in early layers, enabling targeted and effective knowledge editing across architectures.

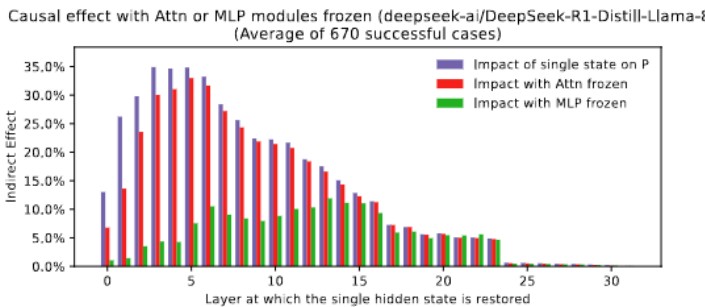

Figure 6: Casual tracing for DeepSeek.

Table 18: DeepSeek's KE performance (%) using MEMIT across editing datasets and different settings of editing layers: GPT2-XL[1] means using GPT2-XL's editing layer settings (Meng et al., 2023), Llama2[2] means using Llama2's editing layer settings (Gupta et al., 2024), CTFC-determined[3] means layers are determined by Casual Tracing with Frozen Components (CTFC) (Meng et al., 2023).

| Editing layer settings | #Edits | zsRE | | | COUNTERFACT | | |
|---|---|---|---|---|---|---|---|
| | | ES | PS | NS | ES | PS | NS |
| GPT2-XL[1] | 0 | 24.69 | 25.24 | 21.21 | 16.00 | 18.00 | 83.20 |
| | $10^2$ | 28.39 | 45.50 | 10..97 | 97.00 | 82.00 | 63.00 |
| | $10^3$ | 18.97 | 23.83 | 3.99 | 93.30 | 73.55 | 61.65 |
| | $10^4$ | 3.53 | 14.92 | 2.02 | 87.09 | 60.70 | 50.70 |
| Llama2[2] | 0 | 24.69 | 25.24 | 21.21 | 16.00 | 18.00 | 83.20 |
| | $10^2$ | 28.69 | 29.57 | 32.06 | 97.00 | 95.00 | 68.50 |
| | $10^3$ | 31.65 | 30.54 | 33.62 | 79.50 | 64.45 | 53.20 |
| | $10^4$ | 0.55 | 0.59 | 2.84 | 65.98 | 56.66 | 48.98 |
| CTFC-determined[3] | 0 | 24.69 | 25.24 | 21.21 | 16.00 | 18.00 | 83.20 |
| | $10^2$ | 31.19 | 31.82 | 31.89 | 86.00 | 66.50 | 56.50 |
| | $10^3$ | 20.17 | 18.89 | 28.26 | 72.20 | 60.90 | 51.01 |
| | $10^4$ | 0.40 | 0.52 | 4.14 | 69.03 | 58.06 | 49.76 |

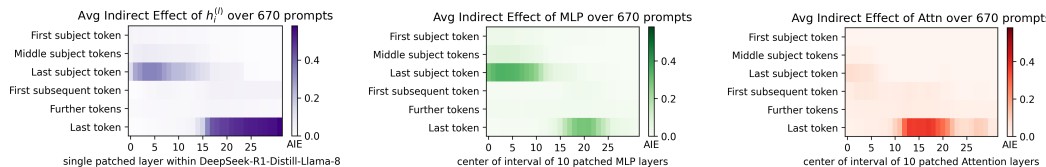

Figure 7: Causal effect heatmaps showing concentrated effects in early layers (1-5) with (from left to right) (a) overall patterns, (b) MLP localisation, and (c) attention mechanisms for the distilled architecture.

We observe that for zsRE dataset with 10,000 edits, all cases perform badly. As shown in Tab. 18, for the case of CTFC-determined, increasing the editing number from 1,000 to 10,000 leads to a dramatic drop in ES, falling from 20.17% to just 0.4%. For cases using Llama2's or GPT's editing layer setting, similar trend also happens: both models' KE performance drops dramatically to a low level (0.55% for LLama2's setting, 3.53% for GPT2-XL's setting) when the editing number rises to 10,000. As shown highlighed by orange, DeepSeeK also perform badly on other KE metrics (i.e. PS, NS) when editing number rises to 10,000, whereas the smallest PS (0.52%) and NS (2.02%) both appears.

### G.3 ANALYSIS

This phenomenon may stem from architectural differences between GPT-series and Llama-based models. In addition, as noted by Wang et al. (2025), discrepancies in pre-training data between GPT-based and Llama-based models can lead to suboptimal simulation of the initial model weights $W_0$, ultimately degrading the effectiveness of KE. Besides, the performance after editing with 10,000 facts is already extremely low, making it highly susceptible to collapsing to near-zero accuracy after fine-tuning. This **instability** prevents meaningful comparison of KE effectiveness before and after fine-tuning on the zsRE dataset. As a result, we do not perform additional fine-tuning experiments on DeepSeek.

# H  DOWNSTREAM-TASK PERFORMANCE

## H.1  ANALYSIS

Table 19: Average and Standard deviation of degradation (%) in evaluation score(Table 6) across models and fine-tuning methods. Avg.[1], Std.[2] are metrics for individual model; Avg_m[3], Std_m[4] are across models.

| Model | Metrics | No ft | Full ft | LoRA | DoRA |
|---|---|---|---|---|---|
| GPT-J | Avg.[1] | 56.84 | 7.17 | 5.19 | 5.87 |
|  | Std.[2] | 17.82 | 12.72 | 8.68 | 4.66 |
| Llama2 | Avg. | -206.92 | 37.27 | 10.81 | 5.93 |
|  | Std. | 220.54 | 8.05 | 15.17 | 8.00 |
| GPT2-XL | Avg. | 1.70 | -0.38 | 2.07 | 8.85 |
|  | Std. | 23.66 | 2.87 | 9.05 | 9.32 |
| | Avg_m[3] | -49.46 | 14.69 | 6.02 | 6.88 |
| | Std_m[4] | 169.81 | 18.6 | 11.63 | 7.50 |

**FT method wise** As shown in Tab. 19, **LoRA exhibits the smallest average performance decrease** ($-6.02\%$) **compared to DoRA** ($-6.88\%$) **and Full fine-tune** ($-14.69\%$). In terms of stability, DoRA demonstrates the lowest standard deviation (7.5) across models, indicating that the impact of KE on DoRA remains relatively consistent regardless of the base model. Notably, models without fine-tuning behave more erratically: the "No ft" group shows the highest average magnitude of change (Avg_m = 49.46%) and standard deviation (Std_m = 169.81), suggesting high variability. This instability may be attributed to base models' relatively poor performances on downstream tasks.

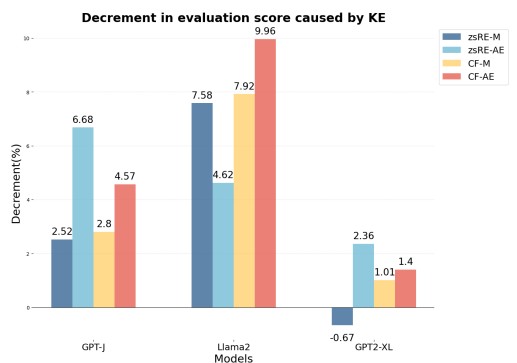

Figure 8: Average decrements ratio (%) caused by KE across models and datasets. M for MEMIT and AE for AlphaEdit. E.g., zsRE-M means MEMIT using zsRE dataset and vice versa.

**Model wise** We found that **GPT2-XL demonstrates the most stable performance across different KE methods and datasets among models evaluated**. As shown in Fig. 8, GPT2-XL has the smallest varying range from $-0.67\%$ to $2.36\%$. In contrast, other models exhibit more variability. For instance, Llama2 experiences a significant fluctuation, with performance ranging from $9.96\%$ to $4.62\%$.

**KE task wise** MEMIT generally leads to a smaller reduction in fine-tuning performance compared to AlphaEdit. As illustrated in Fig. 8, AlphaEdit has the largest average decrements of $9.96\%$ which appears on Llama2. A similar pattern is also observed for GPT-J, where AlphaEdit show larger decrements ($6.68\%$ vs $2.52\%$, and $4.57\%$ vs $2.8\%$) across both datasets.

**COUNTERFACT dataset tends to cause more severe drops in fine-tuning performance**, as evidenced by the top two largest performance declines in Fig. 8 ($9.96\%$, $7.92\%$) occurring in cases involving COUNTERFACT as KE dataset.

**Number of edits also introduces a high degree of variability**, with different patterns observed across models, as detailed in Tab. 6. For example, GPT-J edited with zsRE using MEMIT shows an improvement in fine-tuning performance from $36.95\%$ to $40.92\%$ as the number of edits increases from 100 to 1000, followed by a slight decline to $39.69\%$ when the number of edits reaches 10,000. Conversely, GPT-J, edited with zsRE and fine-tuned with the full dataset, shows a reverse trend, with performance dropping from $35.75\%$ to $33.46\%$ and then rising back to $35.5\%$ as the editing number increases. In addition to these curve-like patterns, a common trend observed in some models is a monotonic decrease in performance. For example, in the case of Llama2 edited by AlphaEdit and fine-tuned by DoRA, the performance consistently drops as the number of edits increases, where performance falls from $78.95\%$ ($10^2$ edits) to $69.61\%$ ($10^3$ edits), and eventually to $63.62\%$ ($10^4$ edits).

## H.2 RESULTS BREAKDOWN

**GPT2-XL** GPT2-XL's performances on downstream tasks after being edited on zsRE dataset are shown in Tab. 20 (no fine-tune and full fine-tune) and Tab. 21 (LoRA and DoRA). Performances of cases being edited on COUNTERFACT dataset are shown in Tab. 22 (no fine-tune and full fine-tune) and Tab. 23 (LoRA and DoRA)

Table 20: Downstream task performances (%) of GPT2-XL ($M_{\text{ed\_ft}}$) edited on zsRE dataset and then being full fine-tuned or not fine-tuned.

| zsRE | No fine-tuning | | | | | | | Full fine-tuning | | | | | | |
| | | MEMIT | | | AlphaEdit | | | | MEMIT | | | MEMIT | | |
| | No edit | $10^2$ | $10^3$ | $10^4$ | $10^2$ | $10^3$ | $10^4$ | No edit | $10^2$ | $10^3$ | $10^4$ | $10^2$ | $10^3$ | $10^4$ |
|---|---|---|---|---|---|---|---|---|---|---|---|---|---|---|
| BoolQ | 58.36 | 59.36 | 58.99 | 48.01 | 59.79 | 59.92 | 24.07 | 6.67 | 9.02 | 14.89 | 14.50 | 6.71 | 6.70 | 5.95 |
| PIQA | 0.80 | 0.65 | 1.80 | 7.40 | 0.76 | 5.06 | 2.01 | 45.16 | 45.16 | 44.18 | 48.53 | 44.89 | 44.23 | 43.47 |
| SIQA | 20.42 | 20.21 | 18.22 | 12.49 | 13.92 | 16.17 | 17.45 | 29.94 | 29.27 | 30.35 | 30.60 | 30.40 | 30.55 | 30.25 |
| HellaSwag | 0.21 | 0.22 | 0.39 | 0.48 | 0.13 | 0.58 | 6.19 | 24.86 | 24.96 | 24.98 | 24.90 | 24.85 | 24.91 | 24.94 |
| WinoGrande | 0.00 | 0.00 | 0.00 | 0.00 | 0.00 | 0.00 | 15.71 | 48.46 | 49.09 | 48.15 | 26.84 | 48.54 | 49.49 | 49.09 |
| ARC-e, | 8.50 | 8.04 | 8.00 | 8.25 | 5.72 | 11.78 | 12.16 | 24.16 | 24.24 | 24.07 | 22.18 | 24.07 | 24.20 | 24.28 |
| ARC-c | 6.32 | 6.66 | 6.06 | 6.91 | 4.18 | 9.90 | 10.41 | 22.35 | 22.27 | 22.44 | 21.59 | 22.35 | 22.18 | 22.44 |
| openbookqa | 12.20 | 13.00 | 12.60 | 14.00 | 9.40 | 16.00 | 17.00 | 27.00 | 27.00 | 26.60 | 24.80 | 26.60 | 26.80 | 26.60 |

Table 21: Downstream task performances (%) of GPT2-XL ($M_{\text{ed\_ft}}$) edited on zsRE dataset and then being LoRA or DoRA fine-tuned.

| zsRE | LoRA | | | | | | | DoRA | | | | | | |
| | | MEMIT | | | AlphaEdit | | | | MEMIT | | | MEMIT | | |
| | No edit | $10^2$ | $10^3$ | $10^4$ | $10^2$ | $10^3$ | $10^4$ | No edit | $10^2$ | $10^3$ | $10^4$ | $10^2$ | $10^3$ | $10^4$ |
|---|---|---|---|---|---|---|---|---|---|---|---|---|---|---|
| BoolQ | 58.72 | 61.75 | 61.87 | 60.04 | 26.00 | 26.96 | 50.09 | 60.80 | 62.42 | 62.42 | 60.43 | 26.18 | 26.76 | 28.50 |
| PIQA | 44.78 | 49.67 | 49.96 | 44.22 | 46.47 | 51.07 | 44.99 | 51.03 | 50.71 | 50.71 | 44.50 | 46.79 | 51.41 | 45.21 |
| SIQA | 39.68 | 42.37 | 38.75 | 37.93 | 19.46 | 38.52 | 32.91 | 39.51 | 39.61 | 39.61 | 37.67 | 19.34 | 38.28 | 35.47 |
| HellaSwag | 24.82 | 25.61 | 25.32 | 25.47 | 25.49 | 25.77 | 25.05 | 22.62 | 25.17 | 25.17 | 25.30 | 25.32 | 25.93 | 21.93 |
| WinoGrande | 49.64 | 50.04 | 49.45 | 41.72 | 51.81 | 50.94 | 48.22 | 48.22 | 49.09 | 49.09 | 41.99 | 52.17 | 50.59 | 46.09 |
| ARC-e, | 21.00 | 27.27 | 27.65 | 25.21 | 16.35 | 24.87 | 24.49 | 25.84 | 27.78 | 27.78 | 25.38 | 16.46 | 25.04 | 24.96 |
| ARC-c | 21.67 | 25.77 | 24.98 | 26.62 | 17.78 | 24.57 | 25.85 | 25.51 | 24.32 | 24.32 | 26.45 | 17.66 | 24.40 | 23.72 |
| openbookqa | 22.00 | 29.40 | 29.20 | 23.00 | 15.00 | 26.80 | 26.20 | 20.00 | 29.60 | 29.60 | 23.40 | 15.00 | 27.00 | 24.60 |

Table 22: Downstream task performances (%) of GPT2-XL ($M_{\text{ed\_ft}}$) edited on COUNTERFACT[1] dataset and then being full fine-tuned or not fine-tuned.

| CF[1] | No fine-tuning | | | | | | | Full fine-tuning | | | | | | |
| | | MEMIT | | | AlphaEdit | | | | MEMIT | | | MEMIT | | |
| | No edit | $10^2$ | $10^3$ | $10^4$ | $10^2$ | $10^3$ | $10^4$ | No edit | $10^2$ | $10^3$ | $10^4$ | $10^2$ | $10^3$ | $10^4$ |
|---|---|---|---|---|---|---|---|---|---|---|---|---|---|---|
| BoolQ | 58.36 | 59.27 | 60.03 | 39.48 | 58.96 | 60.7 | 12.60 | 6.67 | 7.03 | 11.68 | 22.72 | 11.16 | 6.39 | 7.19 |
| PIQA | 0.80 | 0.65 | 0.60 | 14.09 | 0.71 | 2.50 | 18.82 | 45.16 | 41.24 | 46.46 | 46.90 | 45.48 | 42.71 | 44.89 |
| SIQA | 20.42 | 18.68 | 10.29 | 24.21 | 20.21 | 16.53 | 1.18 | 29.94 | 29.48 | 31.42 | 31.83 | 30.45 | 30.19 | 30.30 |
| HellaSwag | 0.21 | 0.33 | 0.38 | 2.47 | 0.14 | 0.97 | 1.74 | 24.86 | 24.83 | 24.92 | 24.85 | 24.75 | 24.85 | 24.93 |
| WinoGrande | 0.00 | 0.00 | 0.00 | 0.00 | 0.00 | 0.16 | 0.24 | 48.46 | 49.25 | 48.93 | 49.25 | 49.09 | 49.17 | 49.57 |
| ARC-e, | 8.50 | 7.32 | 6.99 | 20.03 | 6.78 | 17.21 | 2.78 | 24.16 | 23.95 | 23.36 | 20.29 | 23.65 | 24.28 | 24.16 |
| ARC-c | 6.32 | 5.63 | 4.69 | 17.06 | 5.72 | 15.10 | 1.62 | 22.35 | 22.44 | 22.10 | 19.20 | 22.18 | 22.27 | 22.27 |
| openbookqa | 12.20 | 11.20 | 11.40 | 24.40 | 10.20 | 20.00 | 1.20 | 27.00 | 26.40 | 25.60 | 25.00 | 26.60 | 26.80 | 26.40 |

Table 23: Downstream task performances (%) of GPT2-XL ($M_{\text{ed\_ft}}$) edited on COUNTERFACT[1] dataset and then being LoRA or DoRA fine-tuned.

| CF[1] | LoRA | | | | | | | DoRA | | | | | | |
| | | MEMIT | | | AlphaEdit | | | | MEMIT | | | MEMIT | | |
| | No edit | $10^2$ | $10^3$ | $10^4$ | $10^2$ | $10^3$ | $10^4$ | No edit | $10^2$ | $10^3$ | $10^4$ | $10^2$ | $10^3$ | $10^4$ |
|---|---|---|---|---|---|---|---|---|---|---|---|---|---|---|
| BoolQ | 58.72 | 35.59 | 40.16 | 46.24 | 53.73 | 52.08 | 52.48 | 60.80 | 35.84 | 40.40 | 46.57 | 43.73 | 41.25 | 11.93 |
| PIQA | 44.78 | 41.34 | 46.24 | 42.21 | 48.86 | 49.89 | 47.01 | 51.03 | 41.62 | 45.92 | 48.00 | 25.90 | 51.14 | 47.88 |
| SIQA | 39.68 | 39.22 | 37.94 | 35.02 | 34.03 | 34.29 | 36.44 | 39.51 | 38.95 | 38.13 | 34.80 | 33.52 | 40.74 | 37.87 |
| HellaSwag | 24.82 | 25.27 | 24.42 | 25.47 | 25.11 | 24.18 | 25.10 | 22.62 | 25.10 | 24.57 | 25.30 | 23.44 | 21.99 | 25.06 |
| WinoGrande | 49.64 | 50.87 | 49.46 | 42.49 | 50.67 | 49.33 | 45.54 | 48.22 | 50.51 | 49.8 | 42.78 | 51.30 | 50.04 | 52.09 |
| ARC-e, | 21.00 | 17.26 | 26.09 | 27.22 | 25.25 | 25.93 | 25.67 | 25.84 | 17.38 | 26.26 | 27.44 | 20.16 | 25.72 | 24.45 |
| ARC-c | 21.67 | 18.56 | 25.31 | 24.83 | 23.81 | 25.68 | 24.49 | 25.51 | 18.43 | 25.09 | 25.00 | 20.31 | 25.77 | 25.94 |
| openbookqa | 22.00 | 14.40 | 24.80 | 28.40 | 27.60 | 25.20 | 27.60 | 20.00 | 14.40 | 24.80 | 28.40 | 22.20 | 22.80 | 28.40 |

**Llama2** Llama2's performances on downstream tasks after being edited on zsRE dataset are shown in Tab. 24 (no fine-tune and full fine-tune) and Tab. 25 (LoRA and DoRA). Performances of cases being edited on COUNTERFACT dataset are shown in Tab. 26 (no fine-tune and full fine-tune) and Tab. 27 (LoRA and DoRA)

Table 24: Downstream task performances (%) of Llama2 ($M_{\text{ed\_ft}}$) edited on zsRE dataset and then being full fine-tuned or not fine-tuned.

| | | No fine-tuning | | | | | | | Full fine-tuning | | | | | |
| | | MEMIT | | | AlphaEdit | | | | MEMIT | | | MEMIT | | |
| zsRE | No edit | $10^2$ | $10^3$ | $10^4$ | $10^2$ | $10^3$ | $10^4$ | No edit | $10^2$ | $10^3$ | $10^4$ | $10^2$ | $10^3$ | $10^4$ |
|---|---|---|---|---|---|---|---|---|---|---|---|---|---|---|
| BoolQ | 10.31 | 4.46 | 57.92 | 55.74 | 6.79 | 8.84 | 48.29 | 62.14 | 62.17 | 61.62 | 62.11 | 61.65 | 61.53 | 62.17 |
| PIQA | 0.16 | 0.11 | 5.06 | 0.20 | 0.16 | 0.11 | 0.22 | 70.84 | 46.46 | 42.17 | 28.07 | 34.28 | 32.64 | 50.22 |
| SIQA | 2.15 | 0.26 | 16.17 | 1.52 | 3.99 | 4.55 | 0.41 | 61.26 | 22.88 | 31.83 | 31.99 | 17.96 | 34.08 | 27.43 |
| HellaSwag | 0.00 | 0.58 | 0.58 | 0.00 | 0.00 | 0.00 | 0.00 | 15.77 | 3.50 | 12.67 | 32.96 | 42.06 | 7.35 | 23.16 |
| WinoGrande | 0.00 | 0.16 | 0.00 | 0.00 | 0.00 | 0.00 | 0.00 | 61.72 | 59.19 | 50.51 | 52.09 | 72.53 | 59.59 | 34.25 |
| ARC-e, | 0.67 | 0.21 | 11.78 | 0.52 | 0.80 | 0.97 | 0.04 | 65.07 | 41.33 | 22.77 | 28.11 | 29.34 | 48.48 | 3.91 |
| ARC-c | 0.43 | 0.43 | 9.90 | 0.24 | 0.43 | 0.43 | 0.09 | 46.59 | 23.04 | 19.11 | 24.66 | 23.38 | 34.04 | 3.16 |
| openbookqa | 0.40 | 0.40 | 16.00 | 0.00 | 0.40 | 1.20 | 0.20 | 53.60 | 27.40 | 27.00 | 24.00 | 29.80 | 46.40 | 3.80 |

Table 25: Downstream task performances (%) of Llama2 ($M_{\text{ed\_ft}}$) edited on zsRE dataset and then being LoRA or DoRA fine-tuned.

| | | LoRA | | | | | | | DoRA | | | | | |
| | | MEMIT | | | AlphaEdit | | | | MEMIT | | | MEMIT | | |
| zsRE | No edit | $10^2$ | $10^3$ | $10^4$ | $10^2$ | $10^3$ | $10^4$ | No edit | $10^2$ | $10^3$ | $10^4$ | $10^2$ | $10^3$ | $10^4$ |
|---|---|---|---|---|---|---|---|---|---|---|---|---|---|---|
| BoolQ | 71.16 | 68.87 | 66.09 | 66.09 | 72.97 | 71.01 | 71.04 | 72.45 | 70.87 | 66.94 | 57.45 | 71.10 | 71.50 | 71.59 |
| PIQA | 83.03 | 82.48 | 76.71 | 76.71 | 83.51 | 82.21 | 81.56 | 83.13 | 82.58 | 76.01 | 67.09 | 83.95 | 83.41 | 82.15 |
| SIQA | 79.02 | 78.61 | 77.58 | 77.58 | 78.66 | 79.58 | 79.73 | 80.25 | 79.35 | 76.56 | 66.83 | 79.84 | 79.84 | 78.97 |
| HellaSwag | 90.27 | 87.50 | 61.79 | 61.79 | 90.01 | 88.35 | 87.33 | 89.94 | 88.26 | 58.93 | 67.91 | 91.11 | 88.31 | 85.82 |
| WinoGrande | 83.31 | 82.00 | 78.22 | 78.22 | 83.5 | 83.27 | 81.85 | 84.37 | 83.53 | 77.35 | 70.37 | 82.48 | 83.35 | 81.69 |
| ARC-e, | 83.71 | 80.39 | 71.55 | 71.55 | 84.85 | 83.25 | 82.28 | 82.49 | 80.41 | 71.59 | 65.12 | 83.63 | 83.80 | 81.61 |
| ARC-c | 67.66 | 65.96 | 57.25 | 57.25 | 69.62 | 69.71 | 66.98 | 68.00 | 66.57 | 57.51 | 49.78 | 68.69 | 68.77 | 68.17 |
| openbookqa | 80.80 | 77.80 | 73.00 | 73.00 | 80.80 | 80.00 | 80.40 | 80.20 | 80.00 | 72.00 | 64.40 | 83.80 | 81.20 | 80.80 |

Table 26: Downstream task performances (%) of Llama2 ($M_{\text{ed\_ft}}$) edited on COUNTERFACT[1] dataset and then being full fine-tuned or not fine-tuned.

| | | No fine-tuning | | | | | | | Full fine-tuning | | | | | |
| | | MEMIT | | | AlphaEdit | | | | MEMIT | | | MEMIT | | |
| CF[1] | No edit | $10^2$ | $10^3$ | $10^4$ | $10^2$ | $10^3$ | $10^4$ | No edit | $10^2$ | $10^3$ | $10^4$ | $10^2$ | $10^3$ | $10^4$ |
|---|---|---|---|---|---|---|---|---|---|---|---|---|---|---|
| BoolQ | 10.31 | 55.29 | 55.79 | 52.84 | 7.68 | 20.92 | 51.07 | 62.14 | 61.74 | 62.08 | 62.11 | 62.14 | 55.69 | 62.14 |
| PIQA | 0.16 | 0.33 | 0.22 | 0.2 | 0.16 | 0.11 | 0.11 | 70.84 | 3.54 | 8.87 | 23.88 | 16.27 | 12.57 | 17.85 |
| SIQA | 2.15 | 4.81 | 0.00 | 0.00 | 2.81 | 2.92 | 0.05 | 61.26 | 49.90 | 56.04 | 26.36 | 33.16 | 35.41 | 32.91 |
| HellaSwag | 0.00 | 0.00 | 0.03 | 0.00 | 0.00 | 0.00 | 0.00 | 15.77 | 7.97 | 12.71 | 39.29 | 35.07 | 14.50 | 8.05 |
| WinoGrande | 0.00 | 0.16 | 0.00 | 0.00 | 0.08 | 0.00 | 0.00 | 61.72 | 51.38 | 54.06 | 57.46 | 45.62 | 15.55 | 47.99 |
| ARC-e, | 0.67 | 0.38 | 0.13 | 0.83 | 0.72 | 0.80 | 0.04 | 65.07 | 55.81 | 38.22 | 23.78 | 21.63 | 39.86 | 26.39 |
| ARC-c | 0.43 | 0.26 | 0.00 | 0.26 | 0.34 | 0.85 | 0.26 | 46.59 | 36.26 | 29.18 | 18.43 | 14.93 | 32.25 | 14.16 |
| openbookqa | 0.40 | 0.00 | 0.00 | 0.00 | 0.20 | 0.60 | 0.00 | 53.60 | 44.20 | 36.60 | 21.80 | 23.40 | 32.20 | 26.80 |

Table 27: Downstream task performances (%) of Llama2 ($M_{\text{ed\_ft}}$) edited on COUNTERFACT[1] dataset and then being LoRA or DoRA fine-tuned.

| | | LoRA | | | | | | | DoRA | | | | | |
| | | MEMIT | | | AlphaEdit | | | | MEMIT | | | MEMIT | | |
| CF[1] | No edit | $10^2$ | $10^3$ | $10^4$ | $10^2$ | $10^3$ | $10^4$ | No edit | $10^2$ | $10^3$ | $10^4$ | $10^2$ | $10^3$ | $10^4$ |
|---|---|---|---|---|---|---|---|---|---|---|---|---|---|---|
| BoolQ | 71.16 | 62.26 | 66.64 | 56.54 | 70.37 | 61.59 | 68.23 | 72.45 | 71.63 | 62.97 | 58.03 | 72.14 | 71.44 | 70.58 |
| PIQA | 83.03 | 81.72 | 77.37 | 69.04 | 83.79 | 41.95 | 81.39 | 83.13 | 81.35 | 75.73 | 67.56 | 83.46 | 82.86 | 82.37 |
| SIQA | 79.02 | 80.09 | 77.69 | 67.98 | 79.32 | 48.52 | 78.71 | 80.25 | 78.52 | 76.44 | 67.44 | 80.40 | 79.79 | 78.97 |
| HellaSwag | 90.27 | 40.50 | 71.14 | 64.32 | 51.07 | 9.41 | 84.70 | 89.94 | 85.39 | 76.71 | 68.20 | 89.77 | 89.00 | 83.71 |
| WinoGrande | 83.31 | 83.58 | 78.30 | 72.26 | 83.35 | 59.91 | 81.22 | 84.37 | 84.02 | 77.19 | 70.90 | 82.72 | 81.93 | 80.90 |
| ARC-e, | 83.71 | 83.84 | 72.77 | 66.01 | 83.38 | 14.02 | 81.36 | 82.49 | 81.40 | 72.05 | 65.47 | 83.54 | 83.33 | 82.03 |
| ARC-c | 67.66 | 68.09 | 57.00 | 48.52 | 68.17 | 13.31 | 67.24 | 68.00 | 66.98 | 58.62 | 49.97 | 68.77 | 68.52 | 68.94 |
| openbookqa | 80.80 | 81.80 | 72.80 | 64.80 | 82.00 | 40.20 | 80.20 | 80.20 | 80.00 | 72.40 | 64.80 | 81.20 | 83.00 | 77.80 |

**Llama3.1** Llama3.1's performances on downstream tasks after being edited on zsRE dataset are shown in Tab. 28 (LoRA and DoRA). Performances of cases being edited on COUNTERFACT dataset are shown in Tab. 29 (LoRA and DoRA)

Table 28: Downstream task performances (%) of Llama3.1 ($M_{\text{ed\_ft}}$) edited on zsRE dataset and then being LoRA or DoRA fine-tuned.

| zsRE | | LoRA | | | | | | | DoRA | | | | | |
| | | MEMIT | | | AlphaEdit | | | | MEMIT | | | MEMIT | | |
| | No edit | $10^2$ | $10^3$ | $10^4$ | $10^2$ | $10^3$ | $10^4$ | No edit | $10^2$ | $10^3$ | $10^4$ | $10^2$ | $10^3$ | $10^4$ |
|---|---|---|---|---|---|---|---|---|---|---|---|---|---|---|
| BoolQ | 74.35 | 68.72 | 59.95 | 61.17 | 70.13 | 61.81 | 63.12 | 73.85 | 67.33 | 61.86 | 63.00 | 69.16 | 66.50 | 67.85 |
| PIQA | 84.87 | 78.08 | 67.29 | 69.11 | 81.07 | 70.53 | 72.16 | 88.81 | 82.39 | 75.39 | 77.04 | 84.37 | 81.40 | 82.68 |
| SIQA | 78.80 | 72.49 | 63.12 | 64.91 | 74.86 | 65.37 | 67.38 | 80.27 | 74.65 | 68.47 | 70.01 | 76.26 | 73.61 | 74.73 |
| HellaSwag | 91.57 | 85.16 | 74.09 | 76.64 | 87.00 | 75.69 | 78.30 | 94.20 | 86.65 | 79.28 | 81.05 | 88.49 | 85.40 | 86.69 |
| WinoGrande | 84.17 | 77.44 | 67.37 | 69.70 | 80.36 | 70.73 | 72.74 | 84.67 | 78.33 | 71.78 | 73.42 | 80.14 | 77.34 | 78.54 |
| ARC-e | 84.17 | 77.44 | 66.69 | 68.65 | 79.94 | 69.55 | 71.15 | 90.10 | 83.19 | 76.13 | 77.77 | 85.59 | 82.61 | 83.88 |
| ARC-c | 71.20 | 65.30 | 56.78 | 58.77 | 67.64 | 59.20 | 60.78 | 78.50 | 72.62 | 66.41 | 67.84 | 74.58 | 71.98 | 73.09 |
| openbookqa | 80.80 | 74.40 | 64.80 | 66.80 | 76.80 | 67.00 | 69.00 | 84.80 | 78.40 | 72.00 | 73.60 | 80.60 | 77.80 | 79.00 |

Table 29: Downstream task performances (%) of Llama3 ($M_{\text{ed\_ft}}$) edited on COUNTERFACT[1] dataset and then being LoRA or DoRA fine-tuned.

| CF[1] | | LoRA | | | | | | | DoRA | | | | | |
| | | MEMIT | | | AlphaEdit | | | | MEMIT | | | MEMIT | | |
| | No edit | $10^2$ | $10^3$ | $10^4$ | $10^2$ | $10^3$ | $10^4$ | No edit | $10^2$ | $10^3$ | $10^4$ | $10^2$ | $10^3$ | $10^4$ |
|---|---|---|---|---|---|---|---|---|---|---|---|---|---|---|
| BoolQ | 74.35 | 68.40 | 60.19 | 62.24 | 70.27 | 61.33 | 64.65 | 73.85 | 67.20 | 62.50 | 65.86 | 69.42 | 67.34 | 68.73 |
| PIQA | 84.87 | 78.08 | 67.12 | 70.27 | 80.63 | 70.96 | 75.79 | 88.81 | 81.71 | 76.80 | 73.54 | 85.36 | 82.80 | 83.65 |
| SIQA | 78.80 | 72.50 | 63.80 | 67.43 | 74.07 | 66.66 | 71.11 | 80.27 | 73.05 | 67.21 | 71.59 | 76.26 | 73.21 | 75.50 |
| HellaSwag | 91.57 | 85.16 | 74.89 | 76.64 | 87.99 | 77.43 | 82.71 | 94.20 | 86.66 | 80.59 | 84.06 | 88.55 | 85.89 | 86.78 |
| WinoGrande | 84.17 | 78.28 | 68.88 | 72.02 | 80.36 | 71.52 | 75.54 | 84.67 | 77.90 | 72.45 | 76.34 | 80.44 | 78.03 | 79.64 |
| ARC-e | 84.17 | 77.44 | 67.38 | 69.70 | 79.96 | 70.37 | 75.96 | 90.10 | 82.89 | 77.92 | 74.60 | 85.60 | 83.03 | 83.89 |
| ARC-c | 71.20 | 65.50 | 57.64 | 60.26 | 67.64 | 60.19 | 64.26 | 78.50 | 72.00 | 65.52 | 69.84 | 73.79 | 70.84 | 72.32 |
| openbookqa | 80.80 | 74.34 | 65.42 | 68.39 | 75.95 | 67.59 | 72.15 | 84.80 | 78.86 | 72.56 | 77.28 | 80.56 | 78.14 | 79.75 |

**GPT-J** GPT-J's performances on downstream tasks after being edited on zsRE dataset are shown in Tab. 30 (no fine-tune and full fine-tune) and Tab. 31 (LoRA and DoRA). Performances of cases being edited on COUNTERFACT dataset are shown in Tab. 32 (no fine-tune and full fine-tune) and Tab. 33 (LoRA and DoRA)

Table 30: Downstream task performances (%) of GPT-J ($M_{\text{ed\_ft}}$) edited on zsRE dataset and then being full fine-tuned or not fine-tuned.

| zsRE | | No fine-tuning | | | | | | | Full fine-tuning | | | | | |
| | | MEMIT | | | AlphaEdit | | | | MEMIT | | | MEMIT | | |
| | No edit | $10^2$ | $10^3$ | $10^4$ | $10^2$ | $10^3$ | $10^4$ | No edit | $10^2$ | $10^3$ | $10^4$ | $10^2$ | $10^3$ | $10^4$ |
|---|---|---|---|---|---|---|---|---|---|---|---|---|---|---|
| BoolQ | 56.57 | 31.04 | 36.18 | 35.50 | 31.80 | 27.95 | 8.29 | 33.61 | 51.01 | 40.49 | 30.00 | 47.00 | 22.69 | 21.47 |
| PIQA | 1.36 | 0.87 | 0.98 | 0.82 | 1.20 | 1.74 | 2.61 | 45.38 | 21.71 | 55.71 | 47.44 | 15.61 | 22.09 | 34.44 |
| SIQA | 0.41 | 0.46 | 0.97 | 0.75 | 0.31 | 0.56 | 2.35 | 46.47 | 48.16 | 47.34 | 43.96 | 48.62 | 46.42 | 28.66 |
| HellaSwag | 0.03 | 0.01 | 0.18 | 0.25 | 0.03 | 0.02 | 0.99 | 18.49 | 19.78 | 27.31 | 29.05 | 22.80 | 8.98 | 23.81 |
| WinoGrande | 30.23 | 0.32 | 0.32 | 0.24 | 0.16 | 0.08 | 0.36 | 51.78 | 48.93 | 47.99 | 52.01 | 35.44 | 49.88 | 19.81 |
| ARC-e, | 1.64 | 1.73 | 2.53 | 2.69 | 1.47 | 2.82 | 4.12 | 42.09 | 43.77 | 43.22 | 34.81 | 36.36 | 39.81 | 23.32 |
| ARC-c | 1.02 | 1.28 | 1.96 | 1.28 | 1.11 | 1.19 | 3.67 | 32.08 | 32.00 | 31.91 | 28.84 | 27.22 | 30.80 | 21.50 |
| openbookqa | 1.20 | 1.40 | 3.00 | 4.20 | 1.20 | 1.40 | 5.40 | 29.40 | 30.20 | 33.40 | 27.40 | 27.00 | 24.00 | 24.80 |

Table 31: Downstream task performances (%) of GPT-J ($M_{\text{ed\_ft}}$) edited on zsRE dataset and then being LoRA or DoRA fine-tuned.

| zsRE | | LoRA | | | | | | | DoRA | | | | | |
| | | MEMIT | | | AlphaEdit | | | | MEMIT | | | MEMIT | | |
| | No edit | $10^2$ | $10^3$ | $10^4$ | $10^2$ | $10^3$ | $10^4$ | No edit | $10^2$ | $10^3$ | $10^4$ | $10^2$ | $10^3$ | $10^4$ |
|---|---|---|---|---|---|---|---|---|---|---|---|---|---|---|
| BoolQ | 63.79 | 64.19 | 63.55 | 63.36 | 63.3 | 62.66 | 61.99 | 63.27 | 63.88 | 64.10 | 62.29 | 63.94 | 62.17 | 63.67 |
| PIQA | 73.61 | 73.88 | 69.64 | 71.27 | 70.73 | 72.03 | 58.54 | 73.94 | 73.07 | 74.65 | 72.47 | 74.16 | 70.40 | 68.99 |
| SIQA | 73.39 | 58.09 | 65.92 | 66.12 | 70.93 | 69.04 | 61.72 | 73.39 | 73.54 | 74.41 | 73.69 | 73.95 | 67.35 | 70.47 |
| HellaSwag | 43.86 | 65.88 | 27.27 | 67.56 | 66.61 | 46.10 | 26.19 | 71.00 | 70.84 | 71.36 | 61.69 | 71.80 | 58.10 | 46.80 |
| WinoGrande | 68.75 | 66.38 | 64.33 | 66.85 | 65.67 | 67.56 | 58.56 | 70.24 | 69.69 | 69.14 | 68.75 | 69.38 | 63.85 | 66.46 |
| ARC-e, | 68.31 | 57.20 | 47.77 | 63.38 | 65.32 | 63.09 | 34.39 | 69.44 | 67.93 | 68.10 | 65.91 | 68.22 | 60.31 | 61.11 |
| ARC-c | 51.79 | 43.94 | 33.70 | 49.49 | 50.43 | 46.76 | 27.99 | 53.07 | 53.07 | 51.96 | 49.06 | 53.75 | 47.44 | 46.50 |
| openbookqa | 70.40 | 56.80 | 65.21 | 63.80 | 61.80 | 65.20 | 50.60 | 68.60 | 64.40 | 68.00 | 66.80 | 65.60 | 57.20 | 62.00 |

Table 32: Downstream task performances (%) of GPT-J ($M_{\text{ed\_ft}}$) edited on COUNTERFACT[1] dataset and then being full fine-tuned or not fine-tuned.

| CF[1] | No fine-tuning | | | | | | | Full fine-tuning | | | | | | |
| | | MEMIT | | | AlphaEdit | | | | MEMIT | | | MEMIT | | |
| | No edit | $10^2$ | $10^3$ | $10^4$ | $10^2$ | $10^3$ | $10^4$ | No edit | $10^2$ | $10^3$ | $10^4$ | $10^2$ | $10^3$ | $10^4$ |
|---|---|---|---|---|---|---|---|---|---|---|---|---|---|---|
| BoolQ | 56.57 | 21.93 | 40.34 | 57.16 | 19.27 | 4.37 | 9.20 | 33.61 | 38.69 | 38.32 | 30.70 | 26.57 | 44.37 | 33.15 |
| PIQA | 1.36 | 1.58 | 0.65 | 13.06 | 1.47 | 1.69 | 0.27 | 45.38 | 38.41 | 50.49 | 46.52 | 37.76 | 44.83 | 47.44 |
| SIQA | 0.41 | 0.36 | 0.90 | 0.46 | 0.67 | 2.10 | 4.09 | 46.47 | 45.75 | 40.53 | 39.15 | 48.11 | 43.40 | 35.06 |
| HellaSwag | 0.03 | 0.01 | 0.33 | 2.68 | 0.03 | 0.41 | 0.07 | 18.49 | 24.72 | 26.84 | 21.51 | 22.66 | 17.55 | 16.54 |
| WinoGrande | 30.23 | 0.16 | 0.32 | 0.00 | 0.39 | 0.08 | 0.00 | 51.78 | 51.38 | 52.01 | 50.83 | 51.14 | 30.54 | 36.86 |
| ARC-e, | 1.64 | 1.73 | 3.07 | 2.61 | 1.94 | 2.99 | 8.84 | 42.09 | 44.57 | 39.52 | 29.84 | 45.20 | 35.23 | 24.03 |
| ARC-c | 1.02 | 1.19 | 2.73 | 1.79 | 1.37 | 3.24 | 7.51 | 32.08 | 32.08 | 27.99 | 24.23 | 34.30 | 25.68 | 28.07 |
| openbookqa | 1.20 | 1.80 | 2.40 | 4.60 | 1.80 | 2.40 | 13.00 | 29.40 | 32.40 | 33.40 | 18.40 | 32.60 | 28.40 | 23.00 |

Table 33: Downstream task performances (%) of GPT-J ($M_{\text{ed\_ft}}$) edited on COUNTERFACT[1] dataset and then being LoRA or DoRA fine-tuned.

| CF[1] | LoRA | | | | | | | DoRA | | | | | | |
| | | MEMIT | | | AlphaEdit | | | | MEMIT | | | MEMIT | | |
| | No edit | $10^2$ | $10^3$ | $10^4$ | $10^2$ | $10^3$ | $10^4$ | No edit | $10^2$ | $10^3$ | $10^4$ | $10^2$ | $10^3$ | $10^4$ |
|---|---|---|---|---|---|---|---|---|---|---|---|---|---|---|
| BoolQ | 63.79 | 63.73 | 62.63 | 63.55 | 63.67 | 64.13 | 63.21 | 63.27 | 64.77 | 60.80 | 62.51 | 63.67 | 63.61 | 63.55 |
| PIQA | 73.61 | 74.32 | 73.72 | 73.00 | 73.18 | 70.51 | 68.61 | 73.94 | 74.27 | 72.57 | 71 | 74.65 | 72.14 | 65.72 |
| SIQA | 73.39 | 73.69 | 71.85 | 73.54 | 73.69 | 74 | 68.78 | 73.39 | 71.24 | 67.35 | 61.98 | 74.21 | 70.52 | 70.01 |
| HellaSwag | 43.86 | 19.94 | 71.68 | 66.25 | 71.92 | 40.97 | 33.79 | 71.00 | 71.95 | 32.08 | 64.8 | 71.95 | 51.81 | 49.20 |
| WinoGrande | 68.75 | 69.77 | 68.59 | 66.38 | 68.11 | 50.2 | 65.04 | 70.24 | 69.06 | 67.88 | 65.59 | 70.80 | 69.22 | 64.09 |
| ARC-e, | 68.31 | 66.79 | 67.05 | 65.95 | 68.35 | 66.41 | 60.06 | 69.44 | 66.46 | 68.31 | 59.26 | 68.86 | 66.41 | 59.72 |
| ARC-c | 51.79 | 52.05 | 51.54 | 49.4 | 51.96 | 50.09 | 45.31 | 53.07 | 50.9 | 50.34 | 47.35 | 51.96 | 48.98 | 43.94 |
| openbookqa | 70.40 | 66.00 | 68.00 | 63.20 | 65.80 | 65.60 | 58.20 | 68.60 | 67.00 | 67.20 | 61.20 | 66.40 | 66.80 | 57.80 |

## H.3 PERFORMANCE ON ANOTHER FINE-TUNING DATASET

To enhance the comprehensiveness of our experiments, we include **HotpotQA** as the fine-tuning dataset. In this section, we evaluate the model's KE performance after fine-tuning on HotpotQA. The hyperparameters for this experiment are consistent with those used in the Commonsense dataset. We then compare the results with the performance of models fine-tuned on the Commonsense dataset. As shown in Tab. 34, the HotpotQA group consistently demonstrates lower KE performance compared to the Commonsense group. Regarding *Efficacy*, the largest performance gap is observed with GPT-J edited with 100 zsRE edits using MEMIT, with a difference of 14.31. The difference may result from the use of suboptimal hyperparameters for this dataset. It could also indicate that knowledge-rich datasets causes greater degradation in KE performance. Further analysis is needed to explore this phenomenon.

Table 34: KE performance (%) after fine-tuning using Commonsense and HotpotQA datasets. CS[1] for Commonsense, HQA[2] for HotpotQA.

| #Edits | KE Method | Model | Dataset | FT Method | ES (HQA) | ES (CS[1]) | PS (HQA[2]) | PS (CS) | NS (HQA) | NS (CS) |
|---|---|---|---|---|---|---|---|---|---|---|
| $10^2$ | MEMIT | Llama2 | zsRE | DoRA | 72.76 | 76.3 | 69.65 | 71.69 | 33.02 | 29.69 |
| $10^4$ | AlphaEdit | Llama2 | CF | LoRA | 68 | 70.18 | 48.4 | 64.32 | 83.67 | 62.35 |
| $10^2$ | MEMIT | GPT-J | zsRE | DoRA | 75.56 | 89.87 | 70.63 | 84.62 | 35.71 | 27.64 |
| $10^3$ | AlphaEdit | GPT-J | CF | LoRA | 100 | 99.7 | 87.7 | 90.9 | 82.6 | 79.58 |
| $10^3$ | MEMIT | GPT2-XL | zsRE | DoRA | 39.84 | 45.18 | 27.83 | 43.62 | 29.15 | 24.35 |
| $10^4$ | AlphaEdit | GPT2-XL | CF | LoRA | 41.56 | 53.91 | 32.19 | 41.94 | 78.15 | 71.81 |

## I BREAKDOWN OF ACTIVATION-RELATED ANALYSIS

The statistical results of layer-wise drift and directional similarity analysis are presented in Tab. 35 and Tab. 36, respectively.

Table 35: Layer-wise drift result breakdown

| Metrics | GPT-2-xl M_ed | GPT-2-xl M_ed_tt | GPT-J M_ed | GPT-J M_ed_tt | Llama2 M_ed | Llama2 M_ed_tt |
|---|---|---|---|---|---|---|
| Max | 0.12 | 0.99 | 0.99 | 0.78 | 1 | 0.3 |
| Min | 0 | 0.14 | 0.16 | 0 | 0 | 0.17 |
| Std | 0.05 | 0.36 | 0.37 | 0.35 | 0.54 | 0.57 |
| Avg | 0.04 | 0.16 | 0.14 | 0.21 | 0.21 | 0.24 |

Table 36: Directional similarity result breakdown

| Metrics | GPT2-XL | | | GPT-J | | | Llama2 | | |
|---|---|---|---|---|---|---|---|---|---|
| | M_ed - M_ft | M_ed_ft - M_ft | M_ed_ft - M_ed | M_ed - M_ft | M_ed_ft - M_ft | M_ed_ft - M_ed | M_ed - M_ft | M_ed_ft - M_ft | M_ed_ft - M_ed |
| Max | 0.04 | 0.76 | 0.15 | 0.4 | 0.62 | 0.63 | 0.2 | 0.68 | 0.24 |
| Min | 0 | 0 | 0 | 0 | 0 | 0 | 0 | 0.44 | 0 |
| Std. | 0 | 0.51 | 0.04 | 0.18 | 0.52 | 0.38 | 0.12 | 0.6 | 0.17 |
| Avg. | 0.02 | 0.29 | 0.05 | 0.15 | 0.12 | 0.2 | 0.08 | 0.09 | 0.07 |

# J    SIGNIFICANCE TEST

We use the t-test to evaluate the statistical significance of different FT methods on KE performance across two dimensions: model-wise and FT method-wise. P-values are computed using paired data, where each pair comprises KE performances of an edited-only model ($M_{ed}$) and its edited-and-fine-tuned counterpart ($M_{ed\_ft}$). In the model-wise dimension, experimental configurations are grouped by model, and for each model, p-values are calculated under various FT methods using paired samples with editing counts ranging from 100 to 10,000. For FT method-wise analysis, configurations are grouped by fine-tuning method, and a p-value is computed for each method group.

**Model wise**    As shown in Figure 9, GPT-2 XL exhibits low p-values under both DoRA and LoRA, indicating that the effect of FT on KE performance is genuine and substantial rather than a product of random variation. For Llama, the impact of full fine-tuning in reducing KE performance is the most pronounced among all the models. In the case of GPT-J, although some configurations yield relatively higher p-values, the majority remain consistently low, pointing to strong effects and suggesting that FT influences GPT-J in a targeted and effective manner.

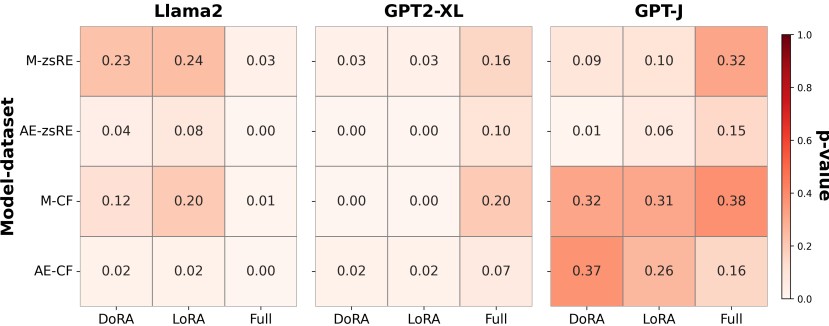

Figure 9: Significance test results across models. The format of Y label is *KE method-KE dataset*, e.g. AE-zsRE means running AlphaEdit on zsRE dataset.

**FT method wise**    The p-values for DoRA, LoRA, and Full Fine-Tuning are $3.29 \times 10^{-5}$, $4.46 \times 10^{-5}$, and 0.009, respectively. These consistently low p-values indicate that the observed performance differences are statistically significant and primarily attributable to the choice of fine-tuning method, rather than intrinsic variation within the editing process itself.

## K  REPRODUCIBILITY RELATED

For KE, we follow exactly the same data process method provided by Meng et al. (2023), and below is an example of hyper-parameters used for Llama3.1. For FT, we follow the Liu et al. (2024)'s work.

```
{
    "alg_name": "MEMIT",
    "model_name": "meta-llama/Llama-3.1-8B",
    "stats_dir": "./data/stats",
    "device": 0,
    "layers": [3, 4, 5, 6, 7],
    "clamp_norm_factor": 4,
    "layer_selection": "all",
    "fact_token": "subject_last",
    "v_num_grad_steps": 25,
    "v_lr": 5e-1,
    "v_loss_layer": 31,
    "v_weight_decay": 1e-3,
    "kl_factor": 0.0625,
    "mom2_adjustment": true,
    "mom2_update_weight": 15000,
    "rewrite_module_tmp": "model.layers.{}.mlp.down_proj",
    "layer_module_tmp": "model.layers.{}",
    "mlp_module_tmp": "model.layers.{}.mlp",
    "attn_module_tmp": "model.layers.{}.self_attn",
    "ln_f_module": "model.norm",
    "lm_head_module": "lm_head",
    "mom2_dataset": "wikipedia",
    "mom2_n_samples": 100000,
    "mom2_dtype": "float32"
}
```

## L  USE OF LARGE LANGUAGE MODELS

In this work, large language models were used for grammar correction and improving the readability of the manuscript. No part of the technical content, including the research design, experimental implementation, data analysis, or interpretation of results, was generated or influenced by an LLM. The role of the model was strictly limited to polishing sentence structure and ensuring clarity in written English.

