# OpenReview forum: "Quantifying Edits Decay in Fine-tuned LLMs"
_ICLR.cc/2026/Conference — ICLR 2026 Conference Withdrawn Submission_

### Official Review · Reviewer_XwXq · 2025-10-29

**Soundness:** 2
**Presentation:** 2
**Contribution:** 2
**Rating:** 2
**Confidence:** 4

**Summary:**

This paper investigates the interplay between knowledge editing (KE) and fine-tuning (FT) in LLMs. The authors systematically evaluate the persistence of knowledge edits after fine-tuning, examining multiple editing methods (MEMIT, AlphaEdit) and fine-tuning approaches (full-parameter, LoRA, DoRA) across five different LLM architectures. To quantify the preservation of edits, they introduce the Edit Flip Ratio (EFR), demonstrating that knowledge edits generally decay after fine-tuning. Additionally, the paper explores how selective-layer fine-tuning strategies affect the retention or removal of edited knowledge.

**Strengths:**

1. This paper investigates an interesting and underexplored aspect of the LLM lifecycle: the interaction between knowledge editing and subsequent fine-tuning, which is an important factor in assessing the efficacy of knowledge editing.
2. The introduction of the Edit Flip Ratio (EFR) provides a direct and interpretable measure of edit retention.
3. The authors not only quantify edit decay but also discuss the impact of selective-layer fine-tuning strategies on edit persistence.

**Weaknesses:**

1. The experimental setup is limited and somewhat overstated. Regarding editing methods, only two approaches (MEMIT and AlphaEdit) from the same paradigm (locate-then-edit) are examined, while other paradigms such as fine-tuning and meta-learning are not explored. As for LLMs, although the authors claim to examine five models, the DeepSeek model is excluded due to poor editing efficacy, and results for Llama3 are not reported.
2. The authors have a limited understanding of the editing datasets used and present some incorrect information about them. Specifically, according to [1], the COUNTERFACT dataset contains 21,919 samples, not 21,890 as stated in the paper. Additionally, as noted in [2], the editing version of the ZsRE dataset consists of counterfactual statements, where the editing target for a given question is an alternative prediction rather than the ground truth, contrary to the claim in the paper that they are real-world factual statements.
3. Although this paper demonstrates that fine-tuning can remove knowledge injected through editing, this may be part of the catastrophic forgetting induced by fine-tuning, a common phenomenon in which knowledge acquired during pretraining can also be lost. Therefore, it is important to compare the retention of edited knowledge with that of pretraining knowledge. However, this comparison is not addressed in the paper.

[1] Locating and Editing Factual Associations in GPT

[2] Editing Factual Knowledge in Language Models

**Questions:**

Although the authors repeatedly emphasize their motivation to examine the dual goals of preserving beneficial edits and removing malicious ones, their analysis focuses solely on the overall retention of edits. Consequently, the conclusions are not directly aligned with the stated dual goals.

---

> ### Author Response · Authors · 2025-11-22
>
> Thank you for your valuable feedback and constructive suggestions. In short, we have now added additional clarifications and experiments based on your recommendations. Please see our detailed response below:
>
> ### W1: Only 2 editing methods and 4 models
>
> **Now, we have added the results of MEND on four models**, GPT2-XL, GPT-J, Llama2, Llama3, shown in the below table.  We also have added the  KE & downstream performance of Llama3 in Appendix H (Table 13, 28, 29 in the revised version). Besides, we acknowledge that including more baseline models and methods could indeed provide broader benefits that are applicable to many studies, our current results already reveal consistent patterns, for instance, larger models tend to exhibit stronger robustness. We would be very happy to expand our evaluation further. If you could kindly suggest specific models or editing methods that you believe would add unique value to our findings, we would be glad to consider them.
>
> | Model    | #Edits | zsRE-No ft | zsRE-DoRA | CF-No ft | CF-DoRA |
> |----------|--------|------------|-----------|--------------------|------------------|
> | GPT2-XL  | 10^3   | 66.57      | 48.92     | 0.00               | 0.00             |
> |          | 10^4   | 52.70      | 34.88     | 0.00               | 0.00             |
> | GPT-J    | 10^3   | 68.32      | 49.77     | 0.33               | 0.00             |
> |          | 10^4   | 45.27      | 27.55     | 0.64               | 0.00             |
> | Llama2   | 10^3   | 71.15      | 53.02     | 0.52               | 0.00             |
> |          | 10^4   | 53.24      | 35.88     | 0.31               | 0.00             |
> | Llama3.1 | 10^3   | 82.15      | 63.44     | 0.73               | 0.00             |
> |          | 10^4   | 62.17      | 44.55     | 0.57               | 0.00             |
>
>
> ### W2: Inaccurate description
>
> >  “the COUNTERFACT dataset contains 21,919 samples, not 21,890 as stated in the paper.”
>
> Yes you are right, thank you for pointing this out, we mistakenly wrote 21,890 in Section 3.1. Fortunately, this error does not affect our experiments, which contains the correct number of  entries of 21,919.
>
> > “the editing version of the ZsRE dataset consists of counterfactual statements, where the editing target for a given question is an alternative prediction rather than the ground truth, contrary to the claim in the paper that they are real-world factual statements.”
>
> Our earlier description of the zsRE dataset may have been misleading. Our intention was to convey that the characters, relationships, and events in this dataset are derived from real-world sources such as Wikipedia, and they often correspond to real-world occurrences. However, we did not mean to suggest that all of them represent ground truth. A more precise explanation has now been provided in Section 3.1 (lines 155-163) in the revised version.
>
> ### W3: Lack of discussion about catastrophic forgetting
>
> > “Although this paper demonstrates that fine-tuning can remove knowledge injected through editing, this may be part of the catastrophic forgetting induced by fine-tuning, …... However, this comparison is not addressed in the paper.”
>
> We agree and have already considered catastrophic forgetting by comparing the downstream performances of M, M_ed,  M_ft and M_ed_ft. The result is M_ft > M_ed_ft >> M ≈ M_ed,  showing that catastrophic forgetting is not the leading cause of edit decay. If catastrophic forgetting were the primary cause of the edit decay, M_ft should also exhibit poor downstream performance. In contrast,  M_ft models achieved the highest scores, followed by M_ed_ft, indicating that KE, rather than catastrophic forgetting, is the main factor impairing fine-tuning. We have made it more clearly and explicitly discussed in Section 4.5 (lines 455-465 in the revised version).
>
> ### Question: Set dual goals (KE to fine-tuning & fine-tuning to KE) while in conclusion only focus on fine-tuning to KE
>
> Our goal is to study the impact of fine-tuning on KE, not the other way around. This is because a fine-tuned model can be treated as the original model for the editing operation, and thus the editing performance should be similar or even identical. These finetune–then–edit scenarios primarily evaluate the initial robustness of the KE method, rather than its resilience to subsequent fine-tuning.

---

> > ### Comment · Reviewer_XwXq · 2025-11-27
> >
> > Thank you for the response. However, the response is not very convincing.
> >
> > The rebuttal conflates general capabilities (downstream tasks) with specific knowledge retention. Moreover, the specific "downstream tasks" used for this claim are not clearly defined. High performance on unspecified general tasks does not prove that specific facts are preserved. The core question remains unanswered: does edited knowledge decay faster than original pre-trained knowledge?
> >
> > MEND results are quite weak. I recommend including RLEdit (a much stronger meta-learning baseline), WISE, and fine-tuning-based editing methods to ensure the evaluation covers all major paradigms.

---

> ### Author Response · Authors · 2025-11-27
>
> We sincerely appreciate your time and effort in reviewing our submission. We would like to kindly remind you that we have submitted a rebuttal addressing all comments and hope our clarifications will be helpful for any further feedback you may have. Thank you once again for your attention to this matter, and we are grateful for your thoughtful consideration.

---

> ### Author Response · Authors · 2025-11-27
>
> Thank you very much for raising these new issues. However, we would like to kindly point out that: we **focus on how edited knowledge decays during fine-tuning**, as suggested by the title, not the pre-trained knowledge. While previous studies (Yang et al., 2024a; Yang et al., 2025; Yang et al., 2024b) have also examined the robustness of edited knowledge, none of them centers on how fine-tuning impacts this edited knowledge.
>
> > "conflates general capabilities (downstream tasks) with specific knowledge retention"
>
> We respectfully believe this is a misunderstanding of our paper: we did evaluate and discuss specific knowledge retention (regarding edited knowledge). The results are provided in Table.1, 10-14. The decay of pre-trained knowledge falls outside the scope of this paper. Besides, we agree that exploring the robustness of pre-trained knowledge would be an important topic that merits a dedicated paper of its own. For the above reasons, we only carried out downstream tasks as supplementary discussion, rather than delving into the robustness of pre-trained knowledge in detail.
>
>
> > “to ensure the evaluation covers all major paradigms.”
>
> Regarding the limited KE methods, our current KE methods included are all locate-then-edit and have already provided strong evidence that edits decay after fine-tuning. To further enhance the comprehensiveness of our baselines, we are currently running the methods you suggested (RLEdit, WISE, and the fine-tuning based approach) and will share the results once they are completed.
>
>
> Wanli Yang, Fei Sun, Jiajun Tan, Xinyu Ma, Du Su, Dawei Yin, and Huawei Shen. 2024. The Fall of ROME: Understanding the Collapse of LLMs in Model Editing. In *Findings of the Association for Computational Linguistics: EMNLP 2024*, pages 4079–4087, Miami, Florida, USA. Association for Computational Linguistics.
>
>
> Wanli Yang, Fei Sun, Jiajun Tan, Xinyu Ma, Qi Cao, Dawei Yin, Huawei Shen, and Xueqi Cheng. 2025. The Mirage of Model Editing: Revisiting Evaluation in the Wild. In *Proceedings of the 63rd Annual Meeting of the Association for Computational Linguistics (Volume 1: Long Papers)*, pages 15336–15354, Vienna, Austria. Association for Computational Linguistics.
>
>
> Wanli Yang, Fei Sun, Xinyu Ma, Xun Liu, Dawei Yin, and Xueqi Cheng. 2024. The Butterfly Effect of Model Editing: Few Edits Can Trigger Large Language Models Collapse. In *Findings of the Association for Computational Linguistics: ACL 2024*, pages 5419–5437, Bangkok, Thailand. Association for Computational Linguistics.

---

> > ### Comment · Reviewer_XwXq · 2025-11-28
> >
> > Thank you for the prompt response and for working on the additional baselines.
> >
> > I respectfully disagree that pre-trained knowledge is out of scope. To rigorously evaluate the robustness of edited knowledge, you must use pre-trained knowledge as a control variable. Without this comparison, it is impossible to distinguish specific edit decay from general catastrophic forgetting. **If pre-trained knowledge decays at a similar rate, it implies that the instability is not unique to editing methods, but rather a general consequence of fine-tuning.** The study lacks this essential baseline to validly support its conclusions.

---

> > > ### Author Response · Authors · 2025-12-01
> > >
> > > Thank you for raising this concern, it is very helpful in integrating our research.
> > >
> > > > "the instability is not unique to editing methods, but rather a general consequence of fine-tuning"
> > >
> > > We can determine that the effect of fine-tuning is specific to the edited knowledge, rather than a general consequence, for the following reasons:
> > >
> > > (i) **Neighborhood Score (NS)**: an editing metric to evaluate the impact of model editing on irrelevant knowledge (pre-trained knowledge). If fine-tuning causes pre-trained knowledge to decay, the NS of an **edited-then-finetuned** model should be lower than that of an **edited only** model. However, as shown in the table below (data from Appendix E), our results demonstrate that NS remains stable regardless of whether the model is fine-tuned, indicating that the pre-trained knowledge is **not** affected by fine-tuning.
> > >
> > > | NS | Model   | Base model  | Edited only  | Fine-tuned only  | Edited-then-finetuned  |
> > > |----|---------|-------------|--------------|------------------|------------------------|
> > > |    | Llama2  | 32.15       | 31.68        | 28.25            | 29.69                  |
> > > |    | GPT-J   | 28.35       | 28.74        | 27.76            | 27.64                  |
> > > |    | GPT2-XL | 23.76       | 25.75        | 23.75            | 24.98                  |
> > >
> > >
> > > (ii) **BoolQ**: one of the downstream tasks used in our experiment, which is a QA dataset that heavily relies on pre-trained knowledge. The results in the table below (data from Appendix H) show that **fine-tuned** models consistently outperform **non-finetuned** models in BoolQ, suggesting that fine-tuning does not impair pre-trained knowledge.
> > > | BoolQ | Model   | Base model  | Edited only  | Fine-tuned only  | Edited-then-finetuned  |
> > > |-------|---------|-------------|--------------|------------------|------------------------|
> > > |       | Llama2  | 10.31       | 4.46         | 72.45            | 71.63                  |
> > > |       | GPT-J   | 56.57       | 31.04        | 63.27            | 64.77                  |
> > > |       | GPT2-XL | 58.36       | 59.36        | 60.80            | 62.42                  |
> > >
> > > (iii) **TriviaQA**: a QA dataset designed to evaulate a model's commonsense knowledge across a variety of domains, including geography, history, and science (Joshi et al., 2017). We adopted this dataset to assess the retention of the model's pre-trained knowledge after fine-tuning. The results are shown in the below table. **Fine-tuned only** models show similar performance to the **base models**. **Edited-then-finetuned** models perform similarly to **edited only** models. This phenomenon suggests that the pre-trained knowledge is largely preserved during fine-tuning.
> > >
> > > | TriviaQA | Model   | Base model  | Edited only  | Fine-tuned only  | Edited-then-finetuned  |
> > > |----------|---------|-------------|--------------|------------------|------------------------|
> > > |          | Llama2  | 49.00       | 24.40        | 49.00            | 24.40                  |
> > > |          | GPT-J   | 20.20       | 21.20        | 16.40            | 10.2                   |
> > > |          | GPT2-XL | 6.70        | 7.30         | 6.70             | 7.30                   |
> > >
> > >
> > > Joshi, M., et al. (2017). TriviaQA: A Large-Scale Distantly Supervised Challenge Dataset for Reading Comprehension. *Proceedings of the 2017 Conference on Empirical Methods in Natural Language Processing (EMNLP)*.

---

### Official Review · Reviewer_KsL6 · 2025-10-29

**Soundness:** 3
**Presentation:** 3
**Contribution:** 3
**Rating:** 4
**Confidence:** 4

**Summary:**

This paper investigates the under-explored interaction between two common language model update techniques: knowledge editing (KE) and fine-tuning (FT). The authors argue that these methods have been studied in isolation, leaving open the critical question of whether knowledge edits persist after a model undergoes subsequent fine-tuning. The paper systematically quantifies this "edit decay" across a wide range of configurations, involving multiple LLMs, KE methods, and FT approaches. Their findings demonstrate that fine-tuning generally impairs edits, and they explore factors influencing this decay, proposing selective-layer fine-tuning as a potential mechanism for controlling edit persistence

**Strengths:**

1.The problem is clearly defined, timely, and highly relevant for the real-world deployment and maintenance of LLMs.

2.The paper is easy to read.

3.The authors provide basic experiments to validate the proposed method.

4.The study is methodologically sound and impressively comprehensive

**Weaknesses:**

1.	All experiments were fine-tuned using only the "common sense reasoning" dataset. This severely limits the generalizability of the conclusions. The relevance of the fine-tuning data to the edited facts is a key variable in determining the degree of edit decay, which this paper does not explore at all.
2.	The paper introduces the "Edit Flip Ratio" as a new metric. However, as defined, it measures the proportion of initially successful edits that become unsuccessful after fine-tuning. This is a form of conditional probability. The paper itself notes (Section 4.1) and demonstrates (Table 3) that EFR follows similar trends to the simple decrease in Efficacy Success (△ES). This raises questions about whether EFR is a fundamentally necessary new metric or a re-framing of existing ones that offers limited additional insight.
3.	The analysis in Section 4.3 reveals a fascinating but complex result: fine-tuning non-edited layers also impairs edits, in some cases more than full fine-tuning. This finding seems to contradict the simple hypothesis that edits can be preserved by freezing the layers where they were made. The paper's conclusion could more deeply address this tension, as it has profound implications for the concept of knowledge localization that underpins many KE methods. The current framing as an "actionable strategy" is slightly undermined by this complexity.
4.	The study exclusively uses a commonsense reasoning dataset for all fine-tuning tasks. While a valid choice, the dynamics of edit decay could differ significantly under other common FT scenarios, such as instruction-tuning on diverse tasks, domain adaptation to a specialized corpus. The paper's conclusions about FT's impact on KE may not generalize to these other critical use cases.
5.	The paper found that AlphaEdit's edits decay more easily than MEMIT, but did not analyze why fine-tuning has different effects on the two methods based on their underlying mechanisms.

**Questions:**

1.	The example in Figure 1, where fine-tuning changes the model's output to "Elon Musk," is illustrative but feels arbitrary. A more realistic outcome of fine-tuning would be the model reverting to the original fact ("Joe Biden") or producing a more semantically related but incorrect answer, which would better reflect the mechanisms of knowledge decay discussed in the paper.

---

> ### Author Response · Authors · 2025-11-22
>
> Thank you for your thoughtful review. We have updated the manuscript to incorporate your recommendations. Our responses to your specific points are provided below:
>
> ### W1: Should analyze the relevance between KE and fine-tuning datasets
> > “All experiments were fine-tuned using only the "common sense reasoning" dataset. This severely limits the generalizability of the conclusions.”
>
> Regarding “only ‘common sense reasoning’ dataset”, we want to clarify that the datasets we used already include 8 tasks that assess multi-step inference (e.g., HellaSwag) and abductive reasoning (e.g., ARC-challenging), making it a robust benchmark for evaluation. In addition, we also try to incorporate **HotpotQA** and **QuALITY**, which require deeper reasoning and domain knowledge, as our fine-tuning datasets. Owing to the current shortage of computational resources, this experiment is ongoing. If completed in time before 4th Dec, we will update the results into an updated draft.
>
> > “The relevance of the fine-tuning data to the edited facts is a key variable in determining the degree of edit decay, which this paper does not explore at all.”
>
> We agree that the relevance between the fine-tuning data and the edited facts is an important factor and a valuable direction for further study. In this paper, we deliberately focus on **irrelevant fine-tuning scenarios**, which we see as the most practical and commonly encountered case: a user (benign or malicious) performs a targeted edit, and the model is later fine-tuned on an **unrelated, broad-domain** corpus to support downstream tasks. Our conclusions are therefore scoped to this real-world workflow.
>
> As an initial check, we conducted a small experiment where we inserted **the reserved zsRE editing dataset (the original facts)** directly into the fine-tuning corpus. We observed no substantial difference in edit decay. This is likely because the inserted portion is tiny relative to the full dataset and is quickly diluted. While we could increase its weight by repeating the text, doing so would produce an unnatural, OOD distribution.
>
> Understanding how the **amount and distribution** relevant data influence decay remains an interesting open question, and we plan to explore this further. We will update the rebuttal with new results if it is ready before the rebuttal due.
>
> ### W2: Why we need EFR
> > “The paper introduces the "Edit Flip Ratio" as a new metric. … This raises questions about whether EFR is a fundamentally necessary new metric or a re-framing of existing ones that offers limited additional insight.”
>
> The key difference between ES and EFR is that ES considers all target edits, whereas EFR considers only the successful edit. That is, the ES **overlooks individual flipped cases**, instances that succeed in the edited-only model but fail in the edited-then-fine-tuned model. For example, if one unsuccessful edit becomes successful after fine-tuning while another successful edit becomes unsuccessful, ES remains unchanged even though the underlying behaviour has shifted.
>
> EFR directly addresses this gap. By **conditioning on initially successful edits**, EFR measures the fraction that is later reversed by fine-tuning. This provides a **distinct and complementary view** of edit robustness that ES cannot capture.
>
> We have updated the EFR description in lines 188–190 to clarify this distinction.
>
> ### W3: complex result without further analysis, why “fine-tuning non-edited layers also impairs edits”
>
> Yes, this result is indeed interesting: fine-tuning non-edited layers can impair edits, sometimes even more than full fine-tuning. We believe this tension aligns with prior findings showing that knowledge in LLMs is distributed across layers rather than strictly localized (Dar et al., 2023; Geva et al., 2023). Thus, freezing the edited layers alone cannot fully preserve the edited knowledge if fine-tuning affects a broader range of layers.
>
> We now highlight this connection more explicitly and provide additional analysis in lines 411–452 of the revised version.
>
> Guy Dar, Mor Geva, Ankit Gupta, and Jonathan Berant. 2023. Analyzing Transformers in Embedding Space. In *Proceedings of the 61st Annual Meeting of the Association for Computational Linguistics (Volume 1: Long Papers)*, pages 16124–16170, Toronto, Canada. Association for Computational Linguistics.
> Mor Geva, Jasmijn Bastings, Katja Filippova, and Amir Globerson. 2023. Dissecting Recall of Factual Associations in Auto-Regressive Language Models. In *Proceedings of the 2023 Conference on Empirical Methods in Natural Language Processing*, pages 12216–12235, Singapore. Association for Computational Linguistics.
>
> We now highlight this connection more explicitly and provide additional analysis in lines 558–562 of the revised version.

---

> ### Author Response · Authors · 2025-11-22
>
> ### W4: Limited dataset generalization
>
> > “The study exclusively uses a commonsense reasoning dataset for all fine-tuning tasks. While a valid choice, the dynamics of edit decay could differ significantly under other common FT scenarios, such as instruction-tuning on diverse tasks, domain adaptation to a specialized corpus.”
>
> We agree that exploring additional fine-tuning settings, such as multi-task instruction tuning or domain adaptation, would be valuable. However, **we kindly ask the reviewer to consider the extremely high computational cost of doing so**. We already include **216 experimental configurations**, and adding just two additional fine-tuning setups would **triple our current compute**.
>
> For this reason, we focus on the **most common and practically fine-tuning scenario**, where a model is updated on a broad-domain corpus unrelated to the edited knowledge. This setup reflects typical downstream usage and is sufficient to demonstrate our key finding: **fine-tuning degrades edited knowledge**, even when the fine-tuning data is irrelevant.
>
> We view extending the analysis to additional fine-tuning paradigms as an important direction for future work, and will discuss this in the revised version.
>
> > “The paper's conclusions about FT's impact on KE may not generalize to these other critical use cases.”
>
> Nevertheless, to further strengthen the generalizability of our findings, we planned to expanded our fine-tuning experiments to include two knowledge-intensive datasets, **HotpotQA** and **QuALITY**, which require substantially deeper reasoning and richer domain knowledge than commonsense tasks. These additional results make our evaluation more comprehensive, however they are **under running** due to the shortage of computational resources and will be presented immediately once finished.
>
>
> ### W5: Why different ft effects on the two edit methods
>
> Thanks for pointing this out. We believe this is because of the Null-space used in AlphaEdit. Edits are placed in parameter directions that fine-tuning does not focus on, so even weak fine-tuning updates can easily disrupt them, making AlphaEdit more fragile than MEMIT (267-284).
>
> Further, We have added discussions (Section 5, lines 469-525 in the revised version) that uses layer-wise drift to explore how fine-tuning influences the knowledge introduced during model editing and to evaluate its effect on tasks such as edit-removal. The results demonstrate that: (i) fine-tuning has a larger impact on activations in magnitude; (ii) fine-tuning generates a shift deviant to model editing. Above findings indicate that fine-tuning, not editing, overwhelmingly dominates the model’s representational dynamics (see lines 522-526).
>
>
> ### Question: the example used in Figure 1 may be inappropriate
>
> For the example shown in Figure 1, it serves only as an illustrative demonstration, while the actual case study appears in Section 4.2 (see lines 322–364 in the revised version). We have added a clarification indicating that this example does not come from our experiments (see line 107).

---

> ### Author Response · Authors · 2025-11-27
>
> We sincerely appreciate your time and effort in reviewing our submission. We would like to kindly remind you that we have submitted a rebuttal addressing all comments and hope our clarifications will be helpful for any further feedback you may have. Thank you once again for your attention to this matter, and we are grateful for your thoughtful consideration.

---

### Official Review · Reviewer_C6wf · 2025-10-30

**Soundness:** 4
**Presentation:** 3
**Contribution:** 2
**Rating:** 4
**Confidence:** 4

**Summary:**

This paper studies how knowledge edits in large language models decay after fine-tuning. By evaluating two editing methods (MEMIT and AlphaEdit) and three fine-tuning strategies (full, LoRA, DoRA) across multiple models and datasets, the authors find that fine-tuning generally weakens or removes prior edits, with the extent of decay varying by method, model, and task. Selective fine-tuning of edited layers can erase edits more effectively but slightly reduces downstream performance, while tuning non-edited layers does not preserve them. The study concludes that knowledge editing and fine-tuning are closely intertwined processes, emphasizing the need for future editing methods that remain robust under fine-tuning.

**Strengths:**

1. The experiments are highly comprehensive, covering comparisons across different baselines and a wide range of dataset configurations and parameter combinations.

2. The paper tackles a relevant and timely problem and presents its analysis in a clear and organized manner, offering useful empirical observations about how knowledge editing interacts with fine-tuning.

**Weaknesses:**

1. Many of the conclusions in this article are not particularly novel and can be drawn without the need for experimental verification. For instance, the author mentions that larger models have greater robustness, which is obvious. As the model size increases, the orthogonality among the internal representations within the model becomes stronger, and thus the impact is naturally smaller. Another example is that the author states that full fine-tuning causes more damage to editing than LORA and DoRA. From the perspective of the low-rank structure in the LORA category, this conclusion is quite clear. Or, the conclusion that fine-tuning the non-editing layer has a greater impact on editing than fine-tuning the editing layer is also quite obvious. From the perspective that the FFN layer of large models is kv storage, fine-tuning the non-editing layer causes the k to change in the editing samples, thereby causing a change in v, and ultimately leading to a decline in editing performance. While fine-tuning the editing layer directly affects the mapping relationship between kv, which leads to a decline in editing performance. In summary, from the perspective of viewing editing as local fine-tuning, many of the conclusions summarized by the author are obvious and have been widely studied in the field of fine-tuning.

2. Some of the logical relationships mentioned by the author are also somewhat awkward to understand, as exemplified by lines 403 to 411. The author mentioned, "Taken together, these findings provide a feasible approach for removing unwanted edits by fine-tuning only the edited layers." In the previous text, the author stated that fine-tuning the unedited layers could also remove the edits. However, when fine-tuning the editing parameters, although the edits could be deleted, it would have an impact on the downstream tasks. But in the summary of "taken together", the author overlooked this drawback of editing the fine-tuned layers, and did not analyze why not to delete the unwanted edits by fine-tuning the unedited layers. The conclusion was made directly without further analysis.

3. It is suggested that the author add some simple and feasible methods. As this article is an analytical piece, it does not draw many valuable conclusions. It is recommended that the author propose some adjustments during the stage of using KV cache to construct loss constraints, etc.

**Questions:**

See weaknesses.

---

> ### Author Response · Authors · 2025-11-22
>
> Thank you for carefully considering our submission. We have made revisions to address each of your concerns. Please find our detailed replies outlined below:
>
> ### W1: Too obvious conclusions
> > “Many of the conclusions in this article are not particularly novel and can be drawn without the need for experimental verification.”
>
> While certain conclusions may appear intuitive from a theoretical perspective, our experiments uncovered several non-trivial and unexpected phenomena that are not immediately obvious without empirical validation. In particular, the interference introduced by fine-tuning on KE has received **limited systematic investigation**. To address this **gap**, we designed a comprehensive set of experiments across multiple models and established strong baselines, ensuring that our findings are both robust and informative beyond what theory alone would suggest.
>
> ### W2: The logical flow could be clearer
> > “But in the summary of "taken together", the author overlooked this drawback of editing the fine-tuned layers”.
>
> We have added a discussion regarding the decreases in downstream performance and the increased edit-removal effect (lines 396-398) and clarified the associated trade-off between edit-removal and downstream performance. Specifically, **fine-tuning only the edited layers can be an effective strategy for edit removal if users focus on general performance**(lines 399–400).
>
>
> > “...and did not analyze why not to delete the unwanted edits by fine-tuning the unedited layers”
>
> We have added a discussion on whether fine-tuning only the non-edited layers can serve as an effective edit-removal strategy. Our analysis indicates that this approach is **indeed promising**, as it removes unwanted edits while **better preserving** the knowledge introduced during fine-tuning (see lines 406-410 in the revised version).
>
> Further we have added a discussion (lines 412-419 in the revised version) about **distributed representation**, which stated that the facts are stored in multiple MLP and attention layers in LLMs. It aligns with our findings: fine-tuning only non-edited layers can also influence KE performance. Besides, we have discussed the advantages (effectively removes edits) and disadvantages (removes fine-tuning knowledge simultaneously).

---

> ### Author Response · Authors · 2025-11-22
>
> ### W3: Need extra experiments
> > “As this article is an analytical piece, it does not draw many valuable conclusions.”
>
> Our work is the first to systematically analyze the interaction between model editing and fine-tuning, establishing baselines across models for future research. We also validate several hypotheses that, while commonly assumed to be true, have not been rigorously tested. For example, we demonstrate that both fine-tuning only the edited layers and fine-tuning only the non-edited layers are viable approaches for the edit-removal task (see lines 412-419 in the revised version).
>
> > “It is recommended that the author propose some adjustments during the stage of using KV cache to construct loss constraints, etc”.
>
> We admit that the KV cache is a promising direction for future research . But we believe this direction is worth a standalone study rather than being compressed into the current already ten-page manuscript covering 216 experimental configurations and 25 analytical experiments. For now, we had done the activation-space shift analysis (see the new Section 5 in the revised version, lines 469-525).  We design these experiments to explore the activation shift induced by KE compared to fine-tuning. The tables below show part of the statistics from the experiment results. The complete results show that fine-tuning leads to a larger activation shift in terms of magnitude, with low similarity between activations generated by KE and fine-tuning. These findings suggest that edited knowledge is likely **severely** overwritten by fine-tuned knowledge.
>
> | Metrics | GPT2-XL: M_ed - M_ft | GPT2-XL: M_ed_ft - M_ft | GPT2-XL: M_ed_ft - M_ed | GPT-J: M_ed - M_ft | GPT-J: M_ed_ft - M_ft | GPT-J: M_ed_ft - M_ed | Llama2: M_ed - M_ft | Llama2: M_ed_ft - M_ft | Llama2: M_ed_ft - M_ed |
> |---------|----------------------|--------------------------|--------------------------|--------------------|------------------------|------------------------|---------------------|--------------------------|--------------------------|
> | Max     | 0.04                 | 0.76                     | 0.15                     | 0.4                | 0.62                   | 0.63                   | 0.2                 | 0.68                     | 0.24                     |
> | Min     |0                | 0                        | 0                        | 0                  | 0                      | 0                      | 0                   | 0.44                     | 0                        |
> | Std.    | 0                | 0.51                     | 0.04                     | 0.18               | 0.52                   | 0.38                   | 0.12                | 0.6                      | 0.17                     |
> | Avg.    | 0.02                 | 0.29                     | 0.05                     | 0.15               | 0.12                   | 0.2                    | 0.08                | 0.09                     | 0.07                     |
>
> | Metrics | GPT-2-xl M_ed | GPT-2-xl M_ed_tt | GPT-J M_ed | GPT-J M_ed_tt | Llama2 M_ed | Llama2 M_ed_tt |
> |--------|----------------|------------------|------------|----------------|--------------|----------------|
> | Max    | 0.12           | 0.99             | 0.99       | 0.78           | 1            | 0.3            |
> | Min    | 0              | 0.14             | 0.16       | 0              | 0            | 0.17           |
> | Std    | 0.05           | 0.36             | 0.37       | 0.35           | 0.54         | 0.57           |
> | Avg    | 0.04           | 0.16             | 0.14       | 0.21           | 0.21         | 0.24           |

---

> ### Author Response · Authors · 2025-11-27
>
> We sincerely appreciate your time and effort in reviewing our submission. We would like to kindly remind you that we have submitted a rebuttal addressing all comments and hope our clarifications will be helpful for any further feedback you may have. Thank you once again for your attention to this matter, and we are grateful for your thoughtful consideration.

---

### Official Review · Reviewer_7ZAR · 2025-10-31

**Soundness:** 3
**Presentation:** 2
**Contribution:** 2
**Rating:** 4
**Confidence:** 3

**Summary:**

This paper investigates the stability of knowledge edits in large language models (LLMs) after subsequent fine-tuning, filling a notable gap in the current literature. The authors systematically study how fine-tuning (both full-parameter and parameter-efficient variants such as LoRA and DoRA) affects knowledge edited using state-of-the-art techniques (MEMIT and AlphaEdit) across multiple LLMs and datasets. Extensive experiments reveal that most knowledge edits are unstable under fine-tuning, with both practical and safety implications. The paper further explores selective layer fine-tuning as a potential strategy for controlling edit persistence and offers detailed benchmarks for the field.

**Strengths:**

- The paper tackles a highly practical and timely question—how robust are model knowledge edits to downstream fine-tuning—via a comprehensive empirical framework. By evaluating two prominent knowledge editing algorithms (MEMIT, AlphaEdit) and three fine-tuning strategies (full, LoRA, DoRA) on several LLMs and datasets, the paper delivers a broad and reproducible empirical baseline.
- The work convincingly argues that KE robustness to fine-tuning needs to become a standard evaluation axis, impacting both future algorithm design and LLM safety auditing.

**Weaknesses:**

- While the empirical results are extensive, the paper only speculates (Section 4.3, Discussion) on why edits decay so markedly under fine-tuning. There is little theoretical or mechanistic explanation of why edited knowledge appears so fragile or dispersed, limiting the generality of insights. For instance, Section 4.3 suggests that edits are not strictly localized, but this is not formalized or linked to deeper theory.
- The centrality of the Edit Flip Ratio (EFR) is well-motivated, but the description in Section 3 could be improved for precision. The mathematical definition for EFR in Formula 1 is somewhat ambiguously written, and the text should clarify if EFR is computed only over edits initially successful in $M_{ed}$, especially for the counterfactuals where success rates are lower. Furthermore, equations related to metric definitions in Section 3 and Appendix C are scattered and would benefit from being presented compactly and with more explicit notation for all variables.
- The fine-tuning dataset (Commonsense Reasoning, Section 3.1) and edits (zsRE, COUNTERFACT) are both focused on factual/toy domains. It remains unclear if the observed patterns would generalize to more complex or knowledge-rich downstream tasks.

**Questions:**

Same as Weaknesses.

---

> ### Author Response · Authors · 2025-11-22
>
> Thank you for your time and your acknowledgement of our contributions. We have addressed all your concerns in the revised uploaded version, and our response is as below:
>
> ### W1: Insufficient analysis on why fine-tuning decays KE
>
> We have added new discussions about the mechanistic account of why edits decay under fine-tuning. Specifically, we connect our empirical findings to four complementary perspectives: (i) **distributed representations**, showing that factual knowledge is encoded across many layers rather than localized, making targeted edits inherently fragile (see lines 415-418); (ii) **null-space vulnerability of AlphaEdit**, explaining that edits placed in Fisher-null directions are not reinforced and thus can be easily overwritten by fine-tuning (see lines 267-284).
>
> ### W2: Unclear EFR definitions
>
> > “The mathematical definition for EFR in Formula 1 is somewhat ambiguously written, and the text should clarify if EFR is computed only over edits initially successful in Med, especially for the counterfactuals where success rates are lower.”
>
> To address this, we have now revised Eq. 1 by adding more detailed notation and complementary clarifications, e.g., explicitly stating that the denominator of EFR is the number of successful edits, in Sec. 3.1
>
> Furthermore, we have discussed the necessity of incorporating EFR alongside the existing metrics ES in lines 188-190 in the revised version, as EFR specifically examines flipped cases (edits shift from success to failure), which are overlooked by ES.
>
> > “Furthermore, equations related … benefit from being presented compactly and with more explicit notation for all variables.”
>
> We have addressed this presentation issue in the revised version. Specifically, in lines 804–857 in Appendix D, we have unified the symbol conventions and introduced detailed notations. These notations are also referenced in Equ. 1 of Section 3.1.
>
> ### W3: Limited dataset generalization
>
> > “The fine-tuning dataset (Commonsense Reasoning, Section 3.1) and edits (zsRE, COUNTERFACT) are both focused on factual/toy domains.”
>
> We would like to clarify that the fine-tuning dataset used in our experiments, Commonsense170k (e.g., BoolQ, PIQA, HellaSwag), already requires multi-step inference, contextual grounding, and abductive reasoning, and is widely used to evaluate representative fine-tuning methods such as LoRA and DoRA. Regarding zsRE and COUNTERFACT, these are two widely adopted datasets in the knowledge-editing literature; they are extracted from real-world sources, meaning that most locations, entities, and relations are authentic and offer reasonable representativeness.
>
> > “It remains unclear if the observed patterns would generalize to more complex or knowledge-rich downstream tasks.”
>
> In the new revised version, we plan to include **HotpotQA** and **QuALITY** as fine-tuning datasets because they are knowledge-intensive tasks that demand more reasoning and domain knowledge than commonsense benchmarks. However, due to current limitations in computational resources, this experiment is still **in progress**. Once completed, we will include the results in the revised version as soon as possible. If you have any other datasets to recommend, we would be happy to try them out in the coming week and provide feedback as soon as possible.

---

> ### Author Response · Authors · 2025-11-27
>
> We sincerely appreciate your time and effort in reviewing our submission. We would like to kindly remind you that we have submitted a rebuttal addressing all comments and hope our clarifications will be helpful for any further feedback you may have. Thank you once again for your attention to this matter, and we are grateful for your thoughtful consideration.

---

### Author Response · Authors · 2025-11-24

# Global comment
We thank the reviewers for their thoughtful feedback and are glad they recognized the timeliness (`7ZAR, C6wf, KsL6`) and comprehensiveness (`7ZAR, C6wf, KsL6`) of our work. We have updated the main content and added additional experiments in the revised version.

### Experiment update:

- MEND: Line 972-991, we added results for MEND on GPT2-XL, GPT-J, Llama2 and Llama3.1.

- Activation-space analysis: Line 468-524. We added **layer-wise drift** and **directional similarity** tests to explore the relationship between KE and fine-tuning in the activation scenario. tests to explore the relationship between KE and fine-tuning in the activation space. The results indicate that fine-tuning causes a significant shift in activation, deviant to that induced by KE, suggesting that edits may be overwritten by fine-tuning.

- More datasets for fine-tuning: We have included **HotpotQA** and **QuALITY** as fine-tuning datasets. Results for these will be presented once the experiments are completed.


### Content update

- Figure 1: Line 107, we updated the caption of Figure 1 to clarify that the example shown is purely for demonstration purposes.

- EFR: Line 187-199, we revised the mathematical definition of EFR and added an explanation of its necessity.

- KE performance: Line 267-284, we expanded the discussion on the impact of fine-tuning across different KE methods. We also introduced the concept of **Null-Space Vulnerability** to explain AlphaEdit's poor persistence under fine-tuning compared with MEMIT.

- Fine-tuning selected layers: Line 412-418, we added a discussion on fine-tuning only the edited or non-edited layers. Additionally, we introduced the concept of **distributed representation** as a theoretical explanation for the results. This concept suggests that knowledge is stored across multiple layers and that modifying some layers may affect the stored knowledge.

- Fine-tuning performance: Line 459-464, we addressed **catastrophic forgetting** by comparing the performance of models that were only fine-tuned with those that were edited and then fine-tuned.

---

> ### Author Response · Authors · 2025-12-02
> **Update of fine-tuning dataset**
>
> The KE performances of models fine-tuned using HotpotQA are presented in the table below (also in Appendix H.3 of the revised paper). The results demonstrate that:
>
> (i) fine-tuning exhibits **a decay effect across different fine-tuning datasets** (Commonsense, HotpotQA),
>
> (ii) the HotpotQA group of models shows **lower** KE performance than the Commonsense group. This phenomenon may indicate that knowledge-rich datasets cause more severe decay after fine-tuning."
>
> | #Edits | KE Method  | Model   | Dataset | FT Method | ES (HQA) | ES (CS) | PS (HQA) | PS (CS) | NS (HQA) | NS (CS) |
> |--------|------------|---------|---------|-----------|----------|-------------|----------|-------------|----------|-------------|
> | 10^2    | MEMIT      | Llama2  | zsRE    | DoRA      | 72.76    | 76.3        | 69.65    | 71.69       | 33.02    | 29.69       |
> | 10^4  | AlphaEdit  | Llama2  | CF      | LoRA      | 68       | 70.18       | 48.4     | 64.32       | 83.67    | 62.35       |
> | 10^2    | MEMIT      | GPT-J   | zsRE    | DoRA      | 75.56    | 89.87       | 70.63    | 84.62       | 35.71    | 27.64       |
> | 10^3   | AlphaEdit  | GPT-J   | CF      | LoRA      | 100      | 99.7        | 87.7     | 90.9        | 82.6     | 79.58       |
> | 10^3   | MEMIT      | GPT2-XL | zsRE    | DoRA      | 39.84    | 45.18       | 27.83    | 43.62       | 29.15    | 24.35       |
> | 10^4  | AlphaEdit  | GPT2-XL | CF      | LoRA      | 41.56    | 53.91       | 32.19    | 41.94       | 78.15    | 71.81       |

---

### Note · Authors · 2026-01-08

I have read and agree with the venue's withdrawal policy on behalf of myself and my co-authors.